# Five New Species of Pezizales from Northeastern China

**DOI:** 10.3390/jof11010060

**Published:** 2025-01-14

**Authors:** Zhengqing Chen, Tolgor Bau

**Affiliations:** 1College of Mycology, Jilin Agricultural University, Changchun 130118, China; czq2324319598@126.com; 2Key Laboratory of Edible Fungal Resources and Utilization (North), Ministry of Agriculture and Rural Affairs, Jilin Agricultural University, Changchun 130118, China

**Keywords:** Pulvinulaceae, Sarcoscyphaceae, phylogeny, taxonomy

## Abstract

Species belonging to the Pezizales are mainly saprobes in nature. They are most commonly observed in woodlands and humid environments. As a result of recent research conducted on the distribution of species in sandy areas and some National Forests Parks, five new species belonging to three genera were identified. A total of five species of disk fungi from Northeast China were identified and described based on morphological classification and molecular phylogenetics. These included *Pulvinula* (*Pulvinula elsenensis*, *Pulvinula sublaeterubra*), *Microstoma* (*Microstoma jilinense*, *Microstoma changchunense*), and *Sarcoscypha* (*Sarcoscypha hongshiensis*). Maximum likelihood and Bayesian analyses were performed using a combined nuc rDNA internal transcribed spacer region (ITS) and nuc 28S rDNA (nrLSU) dataset for the construction of phylogenetic trees. Morphological descriptions, line illustrations, and photographs of the ascocarps of these new species are provided, along with lists of the salient attributes exhibited by the species in the three genera under consideration.

## 1. Introduction

In 1885, Boundier discovered a novel genus, *Pulvinula* Boundier, within the family Humaries. It was characterized by a smooth hymenium, spherical ascospores, and paraphyses that were branched and curved at the apex. The genus initially included *Peziza convexella* Karst., *Peziza sanguinaria* Cooke, and *Peziza constellatio* (Berk. and Br.) [1]. Boundier (1907) later revised the diagnosis of the genus and the nomenclature of two of these species [2]. *Pulvinula* is composed of five species: *Pulvimila cinnabarina* (Fuck.) Bond., *P. carhonaria* (Fuck.) Bond., *P. constellatio* (Berk. and Br.) Bond., *P. hacmastigma* (Iledw. ex Fr.) Bond, and *P. snhanrantia* (Bounn. and Rouss.) Bond. However, this genus was not recognized until 1953, when it was discovered by Le Gal, who used *Peziza convexella* as the species type [3]. Species of this genus are distributed among *Barlaea* Sacc., *Barlaeina* Sacc., *Crouania* Fuckel, *Detonia* Sacc., and *Lamprospora* De Not. [4]. Dennis (1968) stated that one of the characteristics of the species within the genus *Pulvinula* is that the base of the ascus is forked [5]. Pfister (1976) challenged the conclusions of Boudier and Le Gal regarding the synonymy of *Pulvinula haemastigma* (Hedw.) Boud. and *Pulvinula convexella*. This was because no specimens of *Pulvinula haemastigma* then existed and the microscopic structure of the species was not sufficiently characterized to allow a definitive determination of synonymy based on a limited number of key features in its line drawing. Furthermore, the dimensions of the ascus and ascospores had not been documented. Concurrently, Pfister examined five varieties and one subspecies from five distinct genera and determined that they all belonged to *Pulvinula convexella*, representing the diversity within a complex species. Pfister elucidated the rationale behind this categorization, citing the color and dimensions of the ascocarp, the size of the asci and ascospores, and the presence or absence of croziers as determining factors [4]. In 1984, Krof and Zhuang Wenying proposed that the ascospores of species within *Pulvinula* should be classified as either spherical or elliptical and suggested the establishment of a new genus to address this discrepancy [6]. Yao Yijian and Spooner posited that, although *Pulvinula* and *Boubovia* Svrček, are characterized by shared taxonomic traits, including apically curved lateral filaments and forked asci at the base of most species, *Pulvinula* is distinguished by its spherical ascospores and thin-walled asci throughout development, whereas the asci of *Boubovia* species are thick-walled at a specific stage of development [7,8]. Hansen (2013) corroborated the paraphyletic status of the family Pyronemataceae Corda by constructing a phylogenetic tree using multigene fragments. Additionally, he posited that *Boubovia*, *Geopyxis* (Pers.) Sacc., *Pseudombrophila* Boud.m and *Pulvinula* are closely related to the Ascodesmidaceae J. Schröt., and Glaziellaceae J.L. Gibson families. It has been proposed that these four genera can be divided into two or more families [9,10,11]. Subsequently, Ekanayaka et al. (2018) conducted further research and classified three genera (*Pulvinula*, *Lazuardia* Rifai, and *Pseudoboubovia:* U. Lindem., M. Vega, B. Perić and R. Tena) that formed a monophyletic and independent branch on the phylogenetic tree as part of Pezizales J. Schröt. incertae sedis with Pyronemataceae. As a result, the family Pulvinulaceae was established, and *Pulvinula* was designated as the genus type [12,13].

Sarcoscyphaceae Le Gal ex Eckblad is a type of saprophytic fungus that grows on plant matter, including branches, stumps, fallen trees, and twigs. These fungi are distributed throughout tropical and temperate regions. Their main characteristic features are as follows: ascocarps consist of apothecia, which are solitary, scattered, or aggregated. These include disks, cups, goblet shapes, and auricles. Some species within this family have hair on the excipulum, either with or without stipes. They are mostly fleshy or corky in texture, with variable colors ranging from white to orange and to gray [14]. One of the common characteristics of this family is that the asci have thick walls and a suboperculum [15].

The ascus apex structure of the family Sarcoscyphaceae exhibits characteristics that are intermediate between those of the operculum and the inoperculum. In 1946, Le Gal designated this operculum type as a suboperculum and, based on this feature, introduced the name of Sarcoscyphacea for the first time. They further subdivided the species into two tribes, although no Latin diagnosis of the family name was provided at that time [16]. In 1953, Krof identified that this family warranted a special taxonomic status within the order Peziales based on his findings regarding morphological and cytological characteristics [17]. Eckblad (1968) provided a Latin diagnosis of Le Gal’s terminology, conferring legitimacy to the nomenclature [18]. In the same year, Rifai (1968) postulated that the ascospores and subascospores in Pezizales should be in the same position, thus establishing Sarcoscyphineae. At that time, Sarcoscyphineae was only thought to comprise Sarcoscyphaceae, which was further divided into two tribes: Sarcoscypheae, with brightly colred apothecia, and Urnuleae, with dark apothecia [19]. Following a detailed examination of cytology, pigment composition, ultrastructure, asci, and ascospore development (Korf, 1970, 1972, 1973), it was concluded that the classification proposed by Rifai (1968) was valid. This classification elevated the two previously identified tribes to the family level (Sarcoscyphaceae and Sarcosomataceae Kobayasi) and provided a clear delineation of the characteristics associated with these two families [20,21,22]. Subsequent scholars concurred with Korf’s classification [23,24,25]. Cabello (1988) employed a numerical classification method to analyze 22 genera and 34 species of Sarcoscyphineae to investigate the genus relationships within the family Sarcoscyphaceae. His findings supported the view that Sarcoscyphaceae can be distinguished from Sarcosomataceae [24]. Phylogenetic trees of Pezizales and Sarcoscyphaceae were established using molecular techniques, including the analysis of the ITS, SSU, and LSU, as well as other genetic data. These results demonstrate that Sarcoscyphaceae is a monophyletic group [26,27]. In their respective studies, researchers concluded that the family comprises 13 genera and approximately 83 species [12,28].

*Microstoma* Bernstein is mainly characterized by deeply cup-shaped or triangular cup-shaped ascocarps, solitary white hairs on the excipulum, gelatinous ectal excipulum, and smooth ascospores. The most marked classification characteristics of the species within this genus are the presence or absence of a connection between the base of the stipe, the presence or absence of rhizomes, and the shape of the top of the hairs, which may be blunt or sharp [29]. The macroscopic morphology of the species in this genus is similar to that of *Cookeina* Kuntze. Consequently, many researchers have concluded that the two genera are closely related. However, the distinguishing characteristic between the two is that the hairs of *Cookeina* are composed of bundles of hyphae that are typically needle-shaped [30,31]. To date, nine species have been identified in this genus. The species type is *Microstoma protractum* (Fr.) Kanouse (https://indexfungorum.org/Names/Names.asp, accessed on 20 November 2024).

The species of *Sarcoscypha* (Fr.) Boud. are characterized by a disk- or cup-shaped apothecium, with hymenia that may be dirty white, yellow, or scarlet, and ascospores that are elliptical to oblong in shape. The notable features of the ascospores include blunt ends, which may be flat or dented, and surface textures, which range from smooth to nearly smooth, or are at times irregularly wrinkled. The number of guttulates within ascospores is an important diagnostic characteristic for species identification within this genus [32,33,34]. Currently, 35 species are known in this genus. The species type is *Sarcoscypha coccinea* (Jacq.) Lambotte (https://indexfungorum.org/Names/Names.asp, accessed on 20 November 2024).

In 1991, Wenying Zhuang collected 14 species from the genus Sarcoscypha in Jiaohe City, which were classified into three distinct taxa: *Sarcoscypha occidentalis* f. *occidentalis* (Schwein.) Sacc., *Sarcoscypha occidentalis* f. *citrina* W.Y. Zhuang, and *Sarcoscypha vassiljevae* Raitv. This study revealed that the ascospores of these species displayed three distinct surface morphologies: smooth, subsmooth, and wrinkled [35]. In a separate effort, Tolgor Bau (2005) cataloged the ascomycetes of Jilin Province, thereby expanding the known distribution of the *Sarcoscypha* genus in Northeast China [36]. Guiying Chang (2006) documented *Sarcoscypha coccinea* during an investigation of the biological resources in the Zuojia Nature Reserve [37]. In a subsequent study, Chuhan Shi (2016) identified specimens collected between 2013 and 2016, along with those already housed in collections, confirming the presence of 60 species of Pezizales in Jilin Province, representing 20 genera across 7 families. This research also led to the discovery of one new genus and four new species for China, and ten new species were recorded in Jilin Province [38]. Tolgor Bau (2017) observed that the species distribution of Sarcoscyphaceae exhibited distinct patterns across different vegetation zones of Changbai Mountain [39].

Northeast China is situated in the heart of Eurasia, and is characterized by a temperate continental climate. Precipitation is primarily concentrated in the summer and autumn months. This region boasts rich vegetation resources, creating an ideal environment for fungal growth. An investigation of the resources of Northeast China’s sandy land and some National Forests Parks was conducted from 2022 to 2024. This revealed that the species richness of Pezizales in Northeast China is high, including five new species. This study provides a comprehensive account of the newly identified species, accompanied by line illustrations and electron microscope images of ascospores. Additionally, it presents a detailed account of the main characteristics of the three genera and species in tabular form.

## 2. Materials and Methods

### 2.1. Morphological Studies

Specimens were collected in Northeast China between June 2022 and September 2024. The ascocarps were documented with an Olympus Tg-6 camera during the field collection process, with the objective of recording the macroscopic characteristics of the ascocarps. The color of fresh ascocarp was described using the color-coding system developed by the German Institute for Quality Assurance and Certification (Reichs-Ausschuss fur Lieferbedingungen und Guetesicherung, https://www.ral-guetezeichen.de/, accessed on 5 October 2024). Then, specimens were dried using silica gel, and the specimens are currently stored in the herbarium Fungarium of Jilin Agricultural University (FJAU). The line illustrations were based on photos of the ascocarps in the field collection. Light microscopy (LM: Olympus CX33) was used to observe the microstructure, the samples were rehydrated in water, and OPLENIC Pro v1.92 was utilized to measure the tissue structures of asci, ascospores, paraphyses, and the excipulum. Additionally, {a/b/c} represent data for the length, width, and Q value of ascospore, which are derived from a ascospores within b ascocarps across c specimens. d − e × f − i represents the minimum–maximum value of the length × width of the ascospores, and Q = j − k represents the minimum–maximum value of the length/width of the ascospores.

In addition, Melzer’s reagent was employed to determine whether the asci and paraphyses at the apex wall were amyloid or not.

### 2.2. Phylogenetic Studies

DNA was extracted from dried specimens using the NuClean PlantGen DNA kit (CWBIO, Beijing, China). In PCR amplification, the primer pairs ITS1F/ITS and LR0R/LR were utilized [40,41,42,43]. The PCR program followed pre-denaturation at 94 °C for 5 min, followed by 94 °C for 30 s, annealing at 58 °C for 30 s, and extension at 72 °C for 1.5 min. This took 33 cycles. The PCR products were purified and sequenced by Sangon Biotech Co., Ltd. (Shanghai, China). The newly generated sequences were deposited in GenBank.

The phylogenetic analysis included the available sequences of Sarcoscyphaceae and *Pulvinula*. Working according to a study by Zeng Ming et al. (2023), *Chorioactis geaster* (Peck) Kupfer ex Eckblad, *Neournula pouchetii* (Berthet and Riousset) Paden, *Boubovia luteola* Svrček, *Boubovia vermiphila* Brumm., and R. Kristiansen were selected as the outgroups [44].

Finally, the analyzed matrix contained 179 ITS sequences and 70 nrLSU sequences, which are listed in Table 1. The alignment was performed using the online Mafft version 7 (https://mafft.cbrc.jp/alignment/server/, accessed on 10 November 2024) and then manually refined and trimmed using MEGA7. ModelFinder was used to select the best-fit model using the AIC criterion [45].

The best-fit model partition model (edge-unlinked) was selected using the AIC criterion with ModelFinder. For Sarcoscyphaceae, according to the AIC criterion, the best-fit model was GTR + F + R4 for ITS and TIM2 + F + I + G4 (ITS) and TIM2 + F + R3 (nrLSU) for ITS-nrLSU. For *Pulvinula*, according to the AIC criterion, the best-fit model was GTR + F + I + G4 (ITS) and TIM2 + F + G4 (nrLSU) for ITS-nrLSU. Maximum likelihood phylogenies were inferred using the IQTREE under the edge-unlinked partition model for 10,000 ultrafast bootstraps and using the Shimodaira–Hasegawa-like approximate likelihood-ratio test. For Sarcoscyphaceae, Bayesian inference phylogenies were inferred using MrBayes 3.2.6 under the GTR+I+G+F model (2 parallel runs, 2,085,100 generations) for ITS and under the GTR+I+G model (2 parallel runs, 1,488,000 generations) for ITS-nrLSU. For *Pulvinula*, they were inferred under the GTR+I+G model (2 parallel runs, 2,114,700 generations), in which the initial 25% of sampled data were discarded as burn-in [46,47,48,49].

**Table 1 jof-11-00060-t001:** Sequence in formation from phylogenetic trees.

Species	Country	Voucher/Strain Number	GenBank No.		References
			ITS	nrLSU	
*Boubovia luteola*	Germany	R.K. 94/05	KX592793	KX592805	[50]
*Boubovia vermiphila*	Germany	R.K. 89/18	KX592804		[50]
*Chorioactis geaster*	USA	ZZ2 FH	AY307935	AY307943	[51]
*Cookeina colensoi*	Mexico	CUP 62500	AF394040		[31]
*Cookeina colensoi*	Australia	DAR 63642	AF394038		[31]
*Cookeina colensoi*	New Zealand	PDD 55306	AF394037		[31]
*Cookeina cremeirosea*	American Samoa	UTC000275474	KU306964		[52]
*Cookeina cremeirosea*	American Samoa	UTC000275475	KU306963		[52]
*Cookeina garethjonesii*	China	HKAS90509	KY094617	MG871315	[30]
*Cookeina garethjonesii*	China	HKAS90513	KY094622	MG871316	[30]
*Cookeina indica*	Thailand	MFLU 20-0548	MT941004		[53]
*Cookeina indica*	China	HKAS 121171	OK170053	OK398387	[44]
*Cookeina insititia*	China	FH Wang sp 2	AF394033		[31]
*Cookeina insititia*	China	HMAS 71942	AF394031		[31]
*Cookeina korfii*	Philippines	CUP-SA-1797	KT893782		[54]
*Cookeina korfii*	Philippines	CUP-SA-2454	KT893781		[54]
*Cookeina sinensis*	China	HKAS 121175	OK170056	OK398385	[44]
*Cookeina sinensis*	Thailand	MFLU 21-0155	OK413269	OK398383	[44]
*Cookeina speciosa*	Malaysia	C TL 6035	AF394018		[31]
*Cookeina speciosa*	Venezuela	FH Iturriaga 1C-D4	AF394011		[31]
*Cookeina speciosa*	Thailand	MFLU 21-0156	OK413270	OK398390	[31]
*Cookeina sulcipes*	Thailand	MFLU 15-2358	KY094620		[30]
*Cookeina tricholoma*	China	MFLU 15-2359	KY094619	MG871317	[12,30]
*Cookeina tricholoma*	Thailand	MFLU 21-0165	OK413279	OK398394	[44]
*Cookeina venezuelae*	Puerto Rico	FH00432502	AF394041		[31]
*Cookeina venezuelae*	Venezuela	FH Iturriaga 6065	AF394044		[31]
*Cookeina venezuelae*	Guadeloupe	FH00432503	AF394042		[31]
*Geodina guanacastensis*	Bahamas	FH	MN096939	MN096940	[55]
*Geodina guanacastensis*	Costa Rica	CUP CA84	MN096938		[55]
*Kompsoscypha chudei*	China	HKAS 107663A	MT907443	MT907444	[53]
*Microstoma aggregatum*	Japan	TNS: F-88858	LC584238		[56]
*Microstoma aggregatum*	Japan	TNS: F-80795	LC584235		[56]
*Microstoma apiculosporum*	Japan	KPM:NC-28117	LC584241		[56]
*Microstoma apiculosporum*	Japan	TNS: F-37021	LC584239		[56]
** *Microstoma changchunense* **	**China**	**FJAU71961**	**PQ497118**	**PQ498790**	**This study**
** *Microstoma changchunense* **	**China**	**FJAU71962**	**PQ498903**	**PQ498797**	**This study**
*Microstoma floccosum*	Mexico	FH K. Griffith (Micro45)	AF394046		[31]
*Microstoma floccosum*	Mexico	FH K. Griffith (Micro46)	AF394045		[31]
*Microstoma macrosporum*	Japan	TNS: F:91415	LC671644		[56]
*Microstoma macrosporum*	Japan	TNS: F-80822	LC584258		[56]
** *Microstoma jilinense* **	**China**	**FJAU71958**	**PQ496899**	**PQ510812**	**This study**
** *Microstoma jilinense* **	**China**	**FJAU71959**	**PQ496900**	**PQ510814**	**This study**
** *Microstoma jilinense* **	**China**	**FJAU71960**	**PQ497023**	**PQ498789**	**This study**
*Microstoma protracta*	Poland	AW001_Pn	MG920536		[57]
*Microstoma protracta*	Poland	AW010_Kr	MG920535		[57]
*Microstoma longipilum*	Japan	TNS: F-60530	LC584252		[56]
*Microstoma longipilum*	Japan	TNS: F-60527	LC584251		[56]
*Microstoma ningshanicum*	China	SXIM20200016	MW718686	MW718687	[58]
*Microstoma radicatum*	China	CFSZ 10,833 I5I4-3	MG845232		[29]
*Microstoma radicatum*	China	CFSZ 10,833 I5I4-1	MG845230		[29]
***Microstoma* sp.**	**China**	**FJAU71963**	**PQ507629**	**PQ498798**	**This study**
*Nanoscypha striatispora*	China	HMAS 61133	U66016		[26]
*Nanoscypha tetraspora*	Puerto Rico	FH 00464570	AF117352	DQ220374	[26,59]
*Nanoscypha aequispora*	Thailand	MFLU 21-0170	OK413284	OK398399	[44]
*Nanoscypha aequispora*	Thailand	MFLU 21-0171	OK413285	OK398400	[44]
*Neournula pouchetii*	USA	MO 205345	KT968605		[60]
*Neournula pouchetii*		TUR-A195798	JX669837		[61]
*Phillipsia carnicolor*	Thailand	DHP-7126 (FH	AF117353	JQ260811	[59]
*Phillipsia carnicolor*	Thailand	MFLU 18-0713	MH602282		[62]
*Phillipsia chinensis*	China	HMAS 76094	AY254710		[63]
*Phillipsia crispata*	Ecuador	T. Læssøe AAU-44801	AF117354		[59]
*Phillipsia crispata*	Ecuador	T. Læssøe AAU-44895a	AF117355	AY945845	[51]
*Phillipsia domingensis*	USA	CO-1864 (NO)	AF117363		[59]
*Phillipsia domingensis*	China	HKAS 121192	OK170062		[44]
*Phillipsia gelatinosa*	Thailand	MFLU 15-2360	KY498595	KY498589	[64]
*Phillipsia gelatinosa*	Thailand	MFLU 16-2956	KY498593		[64]
*Phillipsia hydei*	Thailand	MFLU 18-0714	MH602283		[62]
*Phillipsia hydei*	Thailand	MFLU 18-1329	MH602284		[62]
*Phillipsia lutea*	French Guiana	NY-4113 (NY)	AF117374	JQ260816	[59]
*Phillipsia olivacea*	Costa Rica	Franco-M 1360 (NY)	AF117375		[59]
*Phillipsia olivacea*	Venezuela	Halling-5456 (NY)	AF117376	JQ260814	[59]
*Phillipsia subpurpurea*	China	MFLU 16-0612	KY498596		[64]
*Pithya cupressina*	USA	mh 208	U66009	JQ260818	[55,65]
*Pithya* sp.	China	DWS8m3	KJ188703		[66]
*Pithya* sp.	USA	T5N32c	AY465469		[67]
*Pithya vulgaris*		RK 90.01	U66008		[65]
*Pithya villosa*	China	HKAS 104653	OK170069	OK398401	[44]
*Pithya villosa*	China	HKAS 121194	OK170068	OK398402	[44]
*Pulvinula archeri*	USA	FDS-CA-03585	PQ211251		[60]
*Pulvinula archeri*	USA	HAY-F-003851	PP789523		[60]
*Pulvinula archeri*	USA	FLAS-F-66445	OR372796	OR360871	[60]
*Pulvinula archeri*	USA	FLAS-F-68939	OR149300	OR134550	[60]
*Pulvinula constellatio*	Italy		AF289074		[68]
*Pulvinula constellatio*	Spain	Ectomycorrhiza	OP847398		[69]
*Pulvinula convexella*	UK	OTU_740s	MT095859		[70]
*Pulvinula convexella*	UK	OTU_739s	MT095858		[70]
** *Pulvinula elsenensis* **	**China**	**FJAU71964**	**PQ507630**		**This study**
** *Pulvinula elsenensis* **	**China**	**FJAU71966**	**PQ507632**	**PQ498805**	**This study**
** *Pulvinula elsenensis* **	**China**	**FJAU71967**	**PQ507631**	**PQ498806**	**This study**
** *Pulvinula elsenensis* **	**China**	**FJAU71968**	**PQ507634**	**PQ498807**	**This study**
** *Pulvinula elsenensis* **	**China**	**FJAU71969**	**PQ507633**	**PQ498808**	**This study**
** *Pulvinula elsenensis* **	**China**	**FJAU71970**	**PQ507635**	**PQ517230**	**This study**
*Pulvinula laeterubra*	USA	FLAS-F-68187	OR149263		[60]
*Pulvinula miltina*	New Zealand	PDD: 106213	OL653010		[60]
*Pulvinula niveoalba*	Spain	ARAN-Fungi 13804	MW248488	MW248512	[71]
*Pulvinula niveoalba*	Germany	M.A.R. 290,809 27	KX592796	KX592808	[50]
*Pulvinula orichalcea*	Spain	ERD-8008	MT432167	MT425197	[60]
*Pulvinula* sp.	Germany		PP461743		[60]
*Pulvinula* sp.	USA	FLAS: F-6890	PP210638		[60]
*Pulvinula* sp.	USA	MES-1034	KY462405		[72]
*Pulvinula* sp.	USA	FLAS-F-70784	OQ150525		[60]
*Pulvinula* sp.	USA	FLAS-F-69785	OQ150476	OP870119	[60]
*Pulvinula* sp.	USA	FLAS-F-69718	OQ150419		[60]
*Pulvinula* sp.	USA	TREC2_d5_C	OL348350		[60]
*Pulvinula* sp.	USA	MICH:352306	OL756006	OL742450	[60]
*Pulvinula* sp.	USA	DBG: F-030576	OP178120		[73]
*Pulvinula* sp.	USA	AK2160	MZ091960	MZ019040	[74]
*Pulvinula* sp.	USA	AEA10751	MZ091959	MZ019039	[74]
*Pulvinula* sp.	USA	FLAS-F-68819	OM672974	OM523302	[60]
*Pulvinula* sp.	USA	FLAS-F-68775	OM672946	OM523282	[60]
*Pulvinula* sp.	USA	FLAS-F-68214	OM672686	OM523218	[60]
*Pulvinula* sp.	Canada	ITS2_OTU_93	MW424606		[75]
*Pulvinula* sp.	UK	OTU_746s	MT095865		[70]
*Pulvinula* sp.	UK	OTU_745s	MT095864		[70]
*Pulvinula* sp.	UK	OTU_744s	MT095863		[70]
*Pulvinula* sp.	UK	OTU_743s	MT095862		[70]
*Pulvinula* sp.	UK	OTU_742s	MT095861		[70]
*Pulvinula* sp.	UK	OTU_741s	MT095860		[70]
*Pulvinula* sp.	UK	OTU_738s	MT095857		[70]
*Pulvinula* sp.	UK	OTU_737s	MT095856		[70]
*Pulvinula* sp.	USA	FLAS-F-61412	MT374021	MT350474	[60]
*Pulvinula* sp.	UK	OTU082s	MK838257		[76]
*Pulvinula* sp.	USA	FLAS-F-63841	MT156532		[60]
*Pulvinula* sp.	USA	OTU1272	MK019168		[60]
*Pulvinula* sp.	USA	OTU992	MK018732		[60]
*Pulvinula* sp.	USA	OTU361	MK018616		[60]
*Pulvinula* sp.	USA	OTU631	MK018575		[60]
*Pulvinula* sp.	USA	OTU336	MK018540		[60]
*Pulvinula* sp.	Mexico	2774	MK397149		[60]
***Pulvinula* sp. 1**	**China**	**FJAU71975**	**PQ507765**	**PQ510821**	**This study**
***Pulvinula* sp. 2**	**China**	**FJAU71976**	**PQ507766**	**PQ510823**	**This study**
** *Pulvinula sublaeterubra* **	**China**	**FJAU71972**	**PQ507726**	**PQ498809**	**This study**
** *Pulvinula sublaeterubra* **	**China**	**FJAU71973**	**PQ507727**	**PQ510830**	**This study**
** *Pulvinula sublaeterubra* **	**China**	**FJAU71974**	**PQ507764**	**PQ498812**	**This study**
** *Pulvinula sublaeterubra* **	**China**	**FJAU71965**	**PQ513528**	**PQ510816**	**This study**
*Rickiella edulis*	Argentina	BAFC 51697	JQ260808		[27]
*Sarcoscypha austriaca*	Norway	CUP 62771	U66010		[65]
*Sarcoscypha austriaca*	USA	CUP 63162	U66011		[65]
*Sarcoscypha coccinea*		AFTOL-ID 50	DQ491486	AY544647	[77]
*Sarcoscypha coccinea*	USA	CUP 63160	U66015		[65]
*Sarcoscypha dudleyi*	USA	CUP 62775	U66018		[65]
*Sarcoscypha dudleyi*	China	HMJAU36044	KU234218		[78]
*Sarcoscypha emarginata*	Luxembourg	CUP 62723	U66020		[65]
*Sarcoscypha emarginata*	China	HB2861	U66021		[65]
** *Sarcoscypha hongshiensis* **	**China**	**FJAU71952**	**PQ496884**	**PQ498771**	**This study**
** *Sarcoscypha hongshiensis* **	**China**	**FJAU71953**	**PQ496886**	**PQ498777**	**This study**
** *Sarcoscypha hongshiensis* **	**China**	**FJAU71954**	**PQ496887**	**PQ498785**	**This study**
** *Sarcoscypha hongshiensis* **	**China**	**FJAU71955**	**PQ496892**	**PQ498784**	**This study**
** *Sarcoscypha hongshiensis* **	**China**	**FJAU71956**	**PQ496893**	**PQ498787**	**This study**
** *Sarcoscypha hongshiensis* **	**China**	**FJAU71957**	**PQ496894**	**PQ498788**	**This study**
*Sarcoscypha hosoyae*		TRL 456	U66031		[65]
*Sarcoscypha humberiana*	China	TNM F28630	KT716833		[79]
*Sarcoscypha humberiana*	China	CUP 63489	U66028		[65]
*Sarcoscypha javensis*	China	HMAS 61198	U66026		[65]
*Sarcoscypha knixoniana*		TRL 1006	U66030		[65]
*Sarcoscypha korfiana*		mh 705	AF026308		[26]
*Sarcoscypha longitudinalis*	China	HKAS 121195	OK170051	OK398403	[44]
*Sarcoscypha longitudinalis*	China	HKAS 121196	OK170052	OK398404	[44]
*Sarcoscypha macaronesica*	Canary Islands	CUP-MM 2628	U66022		[65]
*Sarcoscypha macaronesica*		TFC-MIC 6460	U66023		[65]
*Sarcpscypha mesocyatha*	China	TNM F3688	KT936558		[79]
*Sarcpscypha mesocyatha*	China	TNM F5134	KT936559		[79]
*Sarcoscypha mesocyatha*	USA	CUP 62699	U66029		[65]
*Sarcoscypha minuta*	China	TNM F28831	KT716834		[79]
*Sarcoscypha occidentalis*	USA	CUP 62777	U66024		[65]
*Sarcoscypha occidentalis*	USA	CUP 63484	U66025		[65]
*Sarcoscypha* sp.	China	HMAS 61202	U66027		[65]
*Sarcoscypha tatakensis*	China	TNM F0754	KT716835		[79]
*Sarcoscypha tatakensis*	China	TNM F0993	KT716836		[79]
*Sarcoscypha vassiljevae*	China	HKAS 89817	MG871302	MG871337	[12]
*Sarcoscypha vassiljevae*	China	HMAS 61210	U66017		[65]
*Wynnea americana*	USA	FH 00445978	MK599141	MK599148	[55]
*Wynnea americana*	USA	HKAS 75484	MG871308		[12]
*Wynnea gigantea*	Brazil	FH s. n.	MK335781	MK335801	[80]
*Wynnea gigantea*	Brazil	FH ACM624	MK335782	MK335802	[80]
*Wynnea macrospora*	China	FH 00445975	MK335784	MK335803	[80]
*Wynnea macrospora*	China	FH 00940720	MK335785		[80]
*Wynnea macrotis*	USA	CUP 2684	MK335786	MK335804	[80]
*Wynnea sparassoides*	USA	NYBG02480090	MK335787	MK335805	[80]

New sequences generated for this study are shown in bold.

## 3. Results

### 3.1. Phylogenetic Analyses

This study generated a total of 47 new sequences, including 24 ITS sequences and 23 nrLSU sequences. These new sequences were uploaded to GenBank. The multilocus dataset (ITS + nrLSU) of *Pulvinula* had an aligned length of 1581 total characters and Sarcoscyphaceae had an aligned length of 1103 (ITS) and 1919 (ITS + nrLSU) total characters, including gaps. Only the topological structures of Bayesian inference are displayed, as the topological structures of ML and BI are very similar. Bayesian posterior probability (PP) values ≥ 0.75 and bootstrap support (BS) values ≥ 95% are indicated on branches (PP/BS) (Figure 1 and Figure 2).

The phylogenetic tree of *Pulvinula* (Figure 1) shows that this genus belongs to a polyphyletic group. *Pulvinula elsenensis* and *Pulvinula* sp. 1 (FJAU71975) form a sister-group relationship with high support (PP/BS = 0.95/97), while *Pulvinula sublaeterubra* and *Pulvinula laeterubra* form a sister-group relationship with very high support (PP/BS = 0.99/100).

The phylogenetic tree of Sarcoscyphaceae (Figure 2 and Figure 3) shows that *Microstoma jilinense* forms a distinct branch with full support (PP/BS = 1/100) in both Figure 2 and Figure 3. In Figure 2, it can be seen that *Microstoma changchunense* is in a sister-group relationship with *Microstoma floccosum* and *Microstoma* sp. (FJAU71963), with strong support (PP/BS = 1/100). In Figure 3, *Microstoma changchunense* forms a sister-group relationship with *Microstoma* sp. (FJAU71963), but with lower support (PP/BS = 0.64/76). Additionally, in Figure 2, *Sarcoscypha hongshiensis* and *Sarcoscypha minuta* form a sister-group relationship with high support (PP/BS = 1/99), which is also strongly supported in Figure 3 (PP/BS = 1/100).

### 3.2. Taxonomy

***Pulvinula elsenensis*** T. Bau et Z. Q. Chen, **sp. nov.**

Figure 4A,B; Figure 5

MycoBank number: 856999

Diagnosis: *Pulvinula elsenensis* is characterized by sessile ascocarp; a yellow orange hymenium; and a conspicuous margin. It forms a crozier at the base of asci, has ascospores of 13.0–16.4 μm, and is curved at the apex without enlarged paraphyses.

Etymology: elsenensis is a Mongolian word for “sandy land”, which means that type specimens are collected in sandy area

Type: CHINA. Inner Mongolia Autonomous Region, Tongliao City, 43°01′ N, 122°44′ E, 363 m, 3 June 2023, Weinan Hou and Tolgor Bau (FJAU71966, H230615, holotypus!). Jilin Province Songyuan City, 44°17′ N, 123°35′ E, 133 m, 25 June 2023, Zhengqing Chen (FJAU71964, CZQ2362503, paratypus!).

Description: Apothecia concave–discoid when young, discoid when mature, conspicuous margin, sessile, 0.2–0.7 cm broad. Hymenium smooth, yellow-orange (RAL2000) when fresh. Receptacle surface cream (RAL9001). Ectal excipulum 52.6–78.3 μm broad, composed of textura angularis, hyaline, outer most cell 6.8–12.5 × 4.2–8.5 μm. Medullary excitulum 30.7–42.0 μm broad, composed of textura intricate, hyphae 2.9–4.5 μm, hyaline. Asci cylindrical, 216–268 × 14.2–16.5 μm, 8-spored, operculate, becoming narrow towards the base, distinctly forked base formed by a crozier, J^−^ in Melzer’s reagent, hyaline. Ascospores {6/2/20}, 13.0–16.4 μm, Q = 1.00 − 1.04, spherical, smooth, a large guttulate, hyaline, uniseriate. Paraphyses filiform, slender, septate, branched, curved at apex without enlarged, 2.2–2.9 μm broad.

Habitat: in summer, it grows in clusters on sandy ground covered with mosses.

Distribution: currently, it is only distributed in China’s Inner Mongolia Autonomous Region and Jilin Province.

Additional specimens examined: CHINA. Inner Mongolia Autonomous Region, Tongliao City, 43°01′ N, 122°44′ E, 363 m, 22 August 2022, Weinan Hou and Tolgor Bau (FJAU71970, H2208133). Same location, 17 July 2023, Weinan Hou and Tolgor Bau (FJAU71968, H2307151); 7 September 2023 Weinan Hou and Tolgor Bau (FJAU71967, H230967); Jilin Province Songyuan City, 25 June 2023, Mu Liu (FJAU71969, lm23062).

Notes: In terms of macroscopic morphology, *Pulvinula elsenensis* is easily confused with *P. anthracobia* T. Schumach., *P. multiguttula* (L.R. Batra) S.C. Kaushal, and *P. orichalcea* (Cooke) Rifai in the wild. However, several distinguishing characteristics set it apart. The asci of *P. anthracobia* (140–170 × 11–13 μm) and its ascospores (10.1–12.2 μm) are both smaller, and this species lacks croziers [81]. In contrast, the asci of *P. multiguttula* are wider (14–19 μm) and contain 5–9 guttulates, while the paraphyses are unbranched [82]. *P. orichalcea* can be differentiated by its receptacle surface, which is covered in white pubescence, and its ascospores, which contain three or more small guttulates [83]. Although *Pulvinula elsenensis* shares similarities with *P. tetraspora* (Hansf.) Rifai in its younger stage, the latter may exhibit an orange coloration when fresh and, in contrast, typically has mature asci containing four spores, with a few containing two or five spores [84]. This distinction makes the two species easily identifiable.

***Pulvinula sublaeterubra* T.** Bau et Z. Q. Chen, **sp. nov.**

Figure 4C,D; Figure 6

MycoBank number: 857000

Diagnosis: *Pulvinula sublaeterubra* is characterized by a sessile ascocarp, a pastel orange hymenium, a yellow-orange receptacle surface, and a conspicuous margin. It forms a crozier at the base of asci, displays ascospores at 14.9–17.1 μm, and is curved at the apex without enlarged paraphyses.

Etymology: the specific epithet “sublaeterubra” refers to the similarity of the type species to *Pulvinula laeterubra*.

Type: CHINA. Inner Mongolia Autonomous Region, Tongliao City, 43°01′ N, 122°44′ E, 363 m, 14 July 2022, Weinan Hou and Tolgor Bau (FJAU71974, H220762, holotypus!). Same location, 2 June 2023, Hong Cheng and Tolgor Bau (FJAU71973, C2023060204, paratypus!).

Description: Apothecia concave initially, matting and discoiding to spreasing with maturation, distinct margins, not elevated, 0.2–0.5 cm broad, sessile. Hymenium smooth, pastel orange (RAL2003) when fresh, red-orange (RAL2001) when dry. Receptacle surface yellow-orange (RAL2000) when fresh. Ectal excipulum 152.1–182.4 μm broad, composed of textura angularis, hyaline, outer most cell 8.5–12.0 × 6.1–9.1 μm. medullary excitulum 56.5–75.6 μm broad, composed of textura intricate, hyphae 1.8–3.2 μm, hyaline. Asci cylindrical, 254–302 × 15.7–20.9 μm, 8-spored, operculate, becoming narrow towards the base, distinctly forked base formed by a crozier, J^−^ in Melzer’s reagent, hyaline. Ascospores {4/2/20}, 14.8–17.1 μm, Q = 1.00 − 1.03, spherical, smooth, a large guttulate, hyaline, uniseriate. Paraphyses filiform, slender, septate, branched, curved at apex without enlarged, 1.8–3.2 μm broad.

Habitat: in summer, it grows in clusters on sandy grasslands in broad-leaved forests and shrubs.

Distribution: it is only distributed in China’s Inner Mongolia Autonomous Region.

Additional specimens examined: CHINA. Inner Mongolia Autonomous Region, Tongliao City, 43°01′ N, 122°44′ E, 363 m, 2 June 2023 Weinan Hou and Tolgor Bau (FJAU71965, H230604). Same location, 19 July 2023, Hong Cheng and Tolgor Bau (FJAU71972, C2371915).

Notes: *Pulvinula sublaeterubra* is morphologically similar to *P. miltina*, although the latter can be distinguished by several key characteristics. *P. miltina* has shorter asci (200–250 × 16–20 μm) with either a single large guttulate or multiple smaller guttulates in the ascospores. Additionally, the apexes of its paraphyses are enlarged and unbranched, and typically grow on calcareous substrates [85]. In contrast, the species in question can be differentiated based on their distinct habitat and morphological traits. When compared to *P. nepalensis*, the latter lacks regular croziers, has enlarged paraphysis apexs, and is predominantly found in burned bamboo forests [86]. *P. pyrophila* (Snyder) Donadini, G. Riousset and Riousset, and *P. salmonicolor* (Seaver) Pfister may also be confused with *P. sublaeterubra* in the field. However, *P. pyrophila* is easily distinguishable by its smaller asci (150–200 × 10–12 μm) and ascospores (7–9 μm), as well as its occurrence on burned land in early spring [87]. In contrast, *P. salmonicolor* has larger, wider asci (20–24 μm), larger ascospores (up to 20 μm), and clavate paraphyses, features which facilitate identification [88].

***Microstoma jilinense*** T. Bau et Z. Q. Chen, **sp. nov.**

Figure 4E,F; Figure 7

MycoBank number: 857001

Diagnosis: *Microstoma jilinense* is characterized by salmon orange hymenium; longer hairs at the edges, which are acute toward the apices; asci of 247–325 × 10.6–14.2 μm; elliptical to cylindrical ascospores (15.9–24.3 × 9.0–14.9 μm) without appurtenance on the ends; and fewer paraphyses.

Etymology: the specific epithet “jilinense” refers to the discovery of a type specimen in Jilin Province, China.

Type: CHINA. Jilin Province, Huadian City, Hongshi National Forest Park, 24 June 2024, 127°08′12″ E, 42°49′57″ N, alt. 498 m, Zhengqing Chen (FJAU71958, Q2462405, holotypus!). Liaoning Province, Benxi City, Guanyin Mountain Park, 4 July 2024, 124°6′ E, 41°17′ N, alt. 488, Weinan Hou (FJAU71960, H24070402, paratypus!)

Description: Apothecia deeply cupulate, 0.3–0.65 cm broad, 0.3–0.7 cm deep, sessile or stipitate. Hymenium salmon orange (RAL2012) when fresh, carmine red (RAL 3002) when dry. Receptacle surface salmon orange (RAL2012) when fresh, covered with hairs which longer and gathered into pointed bundles. Stipe 0.5–1.2 × 0.1–0.4 cm, pure white (RAL9010), densely covered with hairs. Ectal excipulum21.5–30.5 μm broad, tissue gelatinous, composed of textura porrecta to textura prismatica, hyaline. Medullary excitulum 84.3–121.3 μm broad, composed of textura intricate, hyphae 2.4–4.1 μm, hyaline. Hairs arising from outer and inner ectal excipular cells, cylindrical, 627–267 μm high, setaceous, septate, acute toward the apices, with thick walls, 1.6–4.6 μm thick. Asci cylindrical, 247–325 × 10.6–14.2 μm, 8-spored, J^−^ in Melzer’s reagent, suboperculate, with slightly thick walls, becoming narrow towards the base. Ascospores {3/1/20}, 15.9–24.3 × 9.0–14.9 μm, Q = 1.43–2.19, elliptical to cylindrical, with a large guttulate when mature, without appurtenance on the ends, smooth, hyaline, uniseriate. Paraphyses filiform, septate, branched, 2.3–4.3 μm broad.

Habitat: in summer, it grows scattered on rotten wood in mixed coniferous and broad-leaved forests.

Distribution: currently, it is only distributed in China’s Jilin Province and Liaoning Province.

Additional specimens examined: CHINA. Jilin Province, Changchun City, Jilin Agricultural University, 22 July 2024, 125°24′19″ E, 43°48′37″ N, alt. 222 m, Tolgor Bau and Zhengqing Chen (FJAU71959, Q247222).

Notes: The ascospores of *Microstoma jilinense* are devoid of appurtenance at both ends, distinguishing it from *M. apiculosporum* (Yei Z. Wan) and *M. camerunense* (Douanla-Meli) [89,90]. Additionally, its more dispersed ascocarp stalk base and the absence of pseudorhizoid differentiate it from *M. protractum* (Fr.) Kanouse and *M. radicatum,* as outlined by T.Z. Liu, Wulantuya, and W.Y. Zhuang [29,91]. Morphologically, *Microstoma jilinense* is often confused with *M. floccosum* (Sacc.) Raitv., which is characterized by a scarlet hymenium and larger ascospores (22–40 × 10–16.8 μm) [15]. Both this species and *M. longipilum* Tochihara, T. Hirao, and Hosoya exhibit dense white hairs along the edge of the hymenium and receptacle that form pointed tufts. However, *M. longipilum* has a dull pink to pale orange hymenium, longer asci (275–350 × 10–17.5 μm), shorter ascospores (11–12.5 μm), and anenlarged apex in some paraphyses [56].

***Microstoma changchunense*** T. Bau et Z. Q. Chen, **sp. nov.**

Figure 4G,H; Figure 8

MycoBank number: 857002

Diagnosis: *Microstoma changchunense* is characterized by light pink hymenium, a cream long stipe, shorter hairs (360–600 μm), asci of 295–345 × 13.5–19.5 μm, and long elliptical ascospores (25.0–36.2 × 11.5–14.5 μm) without appurtenance on the ends.

Etymology: the specific epithet “changchunense” refers to the discovery of a type specimen in Changchun City Jilin Province, China.

Type: CHINA, Jilin Province, Changchun City, Jingyuetan National Forest Park, 9 June 2024, 125°28′01″ E, 43°46′09″ N, alt. 264 m, Tolgor Bau and Yu Wang (FJAU71961, WY24060905, holotypus!); Same location, 9 June 2024, 125°28′01″ E, 43°46′09″ N, alt. 264 m, Tolgor Bau and Tianyu Zhang (FJAU71962, ZTY24695, paratypus!)

Description: Apothecia cupulate, 0.2–0.4 cm broad, 0.4–0.7 cm deep, stipitate. Hymenium light pink (RAL3015) when fresh, red-orange (RAL2001) when dry. Receptacle surface light pink (RAL3015) when fresh, carmine (red RAL3002) when dry, densely covered with hairs at the margins. Stipe 1.9–3.5 × 0.1–0.2 cm, cream (RAL9001). Ectal excipulum 39.8–64.3 μm broad, tissue gelatinous, composed of textura prismatica, hyaline. Medullary excitulum 54.6–85.8 μm broad, composed of textura intricate, hyphae 2.1–3.7 μm, hyaline. Hairs arising from outer and inner ectal excipular cells, cylindrical, 360–600 μm high, setaceous, septate, acute toward the apices, with thickwalls, 3.2–5.7 μm thick. Asci cylindrical, 295–345 × 13.5–19.5 μm, 8-spored, J^−^ in Melzer’s reagent, suboperculate, with slightly thick walls, becoming narrow towards the base. Ascospores {2/2/20}, 25.0–36.2 × 11.5–14.5 μm, Q = 1.90–2.62, long elliptical, with a or more guttulate when mature, without appurtenance on the ends, smooth, hyaline, uniseriate. Paraphyses filiform, septate, branched, 2.1–4.2 μm broad.

Habitat: in summer, it grows scattered on rotten wood in broad-leaved forests.

Distribution: currently, it is only distributed in China’s Jilin Province.

Notes: *Microstoma changchunense* is characterized by a deep, cup-shaped ascocarp and relatively short hairs. It bears resemblance to *M. floccosum*, but the latter is distinguishable by its red hymenium and receptacle [29]. Additionally, while *M. changchunense* shares a similar hymenium and receptacle coloration with *M. macrosporum* (Y. Otani), as outlined by Y. Harada and S. Kudo, *M. macrosporum* differs in having an inverted triangular cup-shaped ascocarp and larger ascospores (42–60 × 16–21 μm) [92]. Misidentification with *M. radicatum* is also possible; however, *M. radicatum* exhibits clustered ascocarps, a dentate margin, and a rhizoid at the base of the stalk [29]. Morphologically, *Microstoma changchunense* resembles *M. ningshanicum,* as outlined by W.Y. Huo, Yu Liu, L.G. Zhang, and J.Z. Li, but its ascospores lack terminal appendages. Furthermore, *M. ningshanicum* has longer asci (449–517 × 16–20.5 μm) and ascospores (31–45 × 13–17 μm), with distinct longitudinal shallow striations, aiding in differentiation [58].

***Sarcoscypha hongshiensis*** T. Bau et Z. Q. Chen, **sp. nov.**

Figure 4I,J; Figure 9

MycoBank number: 857003

Diagnosis: *Sarcoscypha hongshiensis* is characterized by pure orange hymenium, a pure white receptacle surface, a pure white stipe, asci of 284–323 × 9.0–9.9 μm, cylindrical to rod-shaped ascospores (19.2–27.0 × 8.2–9.7 μm) with blunt ends, and both ends being partly truncated.

Etymology: the specific epithet “hongshiensis” refers to the locality from where the type species was collected.

Type: CHINA. Jilin Province, Jiaohe City, Qianjin Forest Farm, 25 August 2023, 127°41′54″ E, 43°57′03″ N, alt. 444 m, Tolgor Bau and Xia Wang, (FJAU71956 W23082515, holotypus!). Jilin Province, Huadian City, Hongshi National Forest Park, 28 August 2023, 127°08′12″ E, 42°49′57″ N, alt. 498 m, Tolgor Bau and Chen zhengqing (FJAU71953, Q2382819, paratypus!).

Description: Apothecia 0.3–2.0 cm broad, discoid, fleshy, stipitate. Hymenium pure orange (RAL2004) when fresh, pastel yellow (RAL1034) when dry, margin integrity. Receptacle surface pure white (RAL9010), with orange-red hue, the edge is similar to hymenium’s color, and longitudinal striations. Stipe subcylindrical, 0.3–4.5 × 0.3–0.5 cm, pure white (RAL9010). Ectal excipulum 66.8–80.6 μm broad, composed of textura porrecta to textura prismatica, cells 17.7–33.9 × 3.7–6.0 μm, hyaline to pale brown. Medullary excitulum 48.9–82.0 μm broad, composed of textura intricate, hyphae 3.5–4.8 μm, hyaline. Asci cylindrical, 284–323 × 9.0–9.9 μm, 8-spored, J^−^ in Melzer’s reagent, suboperculate, becoming narrow towards the base, slim, curvaceous. Ascospores {6/2/20}, 19.2–27.0 × 8.2–9.7 μm, Q = 2.04–3.36, cylindrical to rod-shaped, with blunt ends, partly both ends truncated, smooth, with a large guttulate, hyaline, uniseriate. Paraphyses filiform, septate, branched, 1.6–2.1 μm broad.

Habitat: in summer, it grows in a scattered way on rotten wood in mixed coniferous and broad-leaved forests.

Distribution: currently, it is only distributed in China’s Jilin Province.

Additional specimens examined: CHINA, Jilin Province, Huadian City, Hongshi National Forest Park, 27 August 2023, Tolgor Bau and Chen zhengqing (FJAU71957, Q2382709). Same location, 28 August 2023, Tolgor Bau and Mu Liu (FJAU71955, lm230849); 28 August 2023, T. Bau and Zhengqing Chen (FJAU71952, Q2382809); 28 August 2023, Tolgor Bau and Zhengqing Chen (FJAU71954, Q2382823)

Notes: *Sarcoscypha hongshiensis* shares a similar ascocarp color with *S. macaronesica* Baral and Korf. However, it is distinguishable by the latter’s dentate margin, larger asci (350 × 15.0 μm), and larger ascospores (22–33 × 9.0–11.5 μm) [32]. When comparing it with *S. dudleyi* (Peck) Baral, *Sarcoscypha hongshiensis* stands apart in that *S. dudleyi* is sessile, has longer asci (290–415 × 7–10 μm), and shorter ascospores (18–24 μm) [79]. *Sarcoscypha hongshiensis* may be confused with *S. javensis* Höhn., *S. knixoniana* F.A. Harr., and *S. occidentalis* (Schwein.) Sacc. in the wild. However, *S. occidentalis* is characterized by a receptacle covered in white pubescence and shorter ascospores (18 × 9 μm), whereas *S. javensis* features smaller asci (230 × 10 μm) and ascospores (23 × 8 μm) [93,94]. *S. knixoniana* has longer asci (ranging from 270 to 350 μm, extending up to 420 μm) and ascospores with deep depressions at both ends, which aids in distinguishing it from *Sarcoscypha hongshiensis* [95].

## 4. Discussion

Comprehensive molecular data for many species of *Pulvinula* are lacking, and the molecular information available for the species type, *P. convexella*, is often contradictory. The majority of species are only represented by data from the internal transcribed spacer (ITS) region, with just a few having additional sequences from other genetic markers, such as the large subunit (LSU), RNA polymerase II (RPB2), and translation elongation factor 1-alpha (TEF1-α). This necessitates more detailed and comprehensive data acquisitions to facilitate further studies on *Pulvinula*. The macroscopic morphology and microscopic structure of numerous species of *Pulvinula* are similar. For example, the ascocarps of species in this genus are relatively small, and the color of the hymenium is mainly divided into two categories: white to milky white or light yellow to orange-red (Table 2). It is common for these species to be confused based on their macroscopic morphology alone. Consequently, the utilization of molecular data and phylogenetic analysis is imperative for the accurate identification of *Pulvinula* species that exhibit similar morphological characteristics. *P. elsenensis* and *P. sublaeterubra* form separate branches in the phylogenetic tree; however, the support for *P. elsenensis* is not very high, suggesting that the phylogenetic position of this branch is unstable. *P. sublaeterubra* and *P. laeterubra* are sister groups. However, notable differences are observed between them. The hymenium of *P. laeterubra* is orange-pink, the asci (180 − 200 × 12 μm) and ascospores (10 μm) are smaller, and the habitats of the two are also different. *P. laeterubra* grows exclusively on burnt humus layers [92].

Zeng Ming’s phylogenetic framework was integrated with sequence data from specimens collected in northeastern China, resulting in the construction of a robust phylogenetic tree based on reliable ITS and LSU sequences [44]. The newly described species forms *S. hongshiensis*, a distinct evolutionary lineage within this tree, with *S. minuta* as its sister group. This is outlined by Yei Z. Wang, Cheng L. Huang, and J.L. Wei. However, differences were observed between the two species’ collection sites and habitats. The former is distributed on decaying wood in mixed coniferous and broadleaf forests in a temperate monsoon climate, whereas the latter thrives in a subtropical monsoon climate, where it grows on husks. *S. minuta* can be differentiated by its orange to cadmium yellow ascocarp color, crenulate margins, smaller asci (180–250 × 7–10 μm), and ascospores (16–20 × 8–10 μm) [79]. Additionally, during the species description within this genus, it was noted that the paraphyses of *S. aestiva* Velen., *S. albovillosa* Rehm, and *S. jurana* (Boud.) Baral exhibited a color change when exposed to iodine, turning blue or green (Table 4). No such discoloration was observed in other species, although this phenomenon was recorded in collections of *S. javensis* and *S. hongshiensis.* Given the limited number of species collected so far, it remains difficult to confirm this phenomenon. Additional specimens of these species are required to validate and elucidate these characteristics.

The species of *Microstoma* are primarily distributed in temperate regions where they either grow in aggregates or as solitary specimens. The receptacle surface is covered with white hairs composed of individual hyphae. This is a key distinguishing feature of the closely related genus *Cookeina*, which is typically solitary and found predominantly in tropical and subtropical regions [30,31,96]. The key morphological traits for species differentiation within *Microstoma* include the color of the ascospore disks, the presence of pseudorhizomes, the shape of the hairs at the edges, the shape of the ascospores, the presence or absence of appurtenance on at ends, and the number of guttulates (Table 3). In the present study, *M. jilinense* and *M. changchunense* were clearly differentiated from other species within the genus (Table 3). This differentiation was supported by the absence of pseudorhizomes in both species, the lack of appurtenance at the ends of the ascospores, and the observation that *M. jilinense* exhibited longer hairs (627–1267 μm) than *M. changchunense*, which had shorter hairs (360–600 μm).

Furthermore, *Microstoma* sp. (FJAU71963), *Pulvinula* sp. 1 (FJAU71975), and *Pulvinula* sp. 2 (FJAU71976) each produced a distinct and separate evolutionary lineage within the phylogenetic tree. However, this particular lineage has not been characterized or described at present, owing to only one specimen being available for observation.

The northeastern region of China has a substantial diversity of Pezizales species, with *Pulvinula*, *Microstoma*, and *Sarcoscypha* being predominant. In the present study, five novel species were identified and characterized based on morphological and phylogenetic analyses, thereby augmenting the species richness of Pezizales in this region. Concurrently, these results imply the potential existence of hitherto undetected species in northeastern China, which merit further in-depth investigation.

**Table 2 jof-11-00060-t002:** Current list of *Pulvinula* species and their diagnostic characteristics. Non-English descriptions from references are not translated to avoid misunderstanding.

Species	Apothecia Color When Fresh	Asci	Crozier	Ascospores	Paraphyses	Habitat	References
*P. alba*	White	Cylindric, 225–290 × 20–26(–30) μm, 8-spored	Base forked	Unicellular, globose, 17.0–19.0(20.5) μm, one large guttule, smooth under the ligth microscope	Filiform, septate, apically curved, not enlarged at the apex	On damp ground	[8]
*P. albida*	Albidis	180 μm longis, 16 μm latis		Globosis, 15 μm,	filiformes	At terram	[97]
*P. anthracobia*	Orange to crimson	Cylindrical, 140–170 × 11–13 μm, 8-spored	Unforked base	Globose, smooth, a single large oil globule, 10.1–12.2 μm	Filiform, 0.9–1.6 μm, moderately curved and branched from the middle of their length, not enlarged or branched towards the apex, slightly longer than the asci	On soil and charred wood in fire bed	[81]
*P. archeri*	Cinnabarina, bright-crimson	Cylindrical		Globes, 1/3500 inch across, with a large uncleus		On dead leaves of some succulent plant	[98]
*P. carbonaria*	aurantio-sanguineis	Cylindraceis, 8sporis, 116 Mik., long, 16 Mik. crass		Golbosis, episporio reticulato, 16Mik., diamater	Filiformibus, aurantiacia,copiosis	Auf verlassenen brandstellen kohlenmeilern und dergl., sehr selten, im Herbst.	[99]
*P. cinnabarina*	Cinnabarinis	Cylindraceis, 8sporis, 144 Mik. long, 18 Mik. crass		Golbosis, episporio reticulato, 18Mik., diamater	Filiformibus, copiosis, aurantiacia	Auf dem sande des Rheinbettes bei Ragaz, in der Schweiz, nachst der Eisenbahnbrucke, hier haufig, im Herbst	[99]
*P. convexella*	Sangvineo-rubra			Sphaeroideae, 15–16 mmm	Graciles, flexuoxae, intus granulosae	supra terram nudam in rivuli margine	[100]
*P. discoidea*	Albo-cinereis	Cylindreeeo-clavatia, apice rotundatis, 8sporis, 100–110 × 14–16 μm		Globosis,lacvibus, hyaline-subtlavesentibus, 12–14 μm	Filiformilms,guttulatis, vel septatis	Auf lehuigem erdboden wachsenden	[101]
*P. elsenensis*	Yellow-orange	cylindrical, 216–268 × 14.2–16.5 μm, 8-spored, operculate, becoming narrow towards the base	distinctly forked base formed by a crozier	spherical, 13.0–16.4 μm, smooth,	filiform, slender, septate, branched, curved at apex without enlarged	grows in clusters on sandy ground covered with mosses	This study
*P. etiolata*	Alba	cylindraceis		Globosis, laevibus	filiformibus	On the ground	[102]
*P. globifera*				Globosis, 0.002 inch		On rutten logs in wood	[103]
*P. johannis*	Pale or more or less deep pink samon	Cylindrical or cylindrical–clavate, 170–200 × 12–14.5 μm, 8-spored	A distinctly forked base formed by a crozier	Spherical, 9–11 μm, generally with a large and sometimes eccentric oil-drop, smooth, uniseriate in the ascus	Very slender, thread-shape, curved or hooked in the upper part simple (not forked), some weakly visible septa	on bare, humid ground	[104]
*P. lacteoalba*	Tota lacteoalba	Cylindracei, tetraspori, 180–230 × 15 μm		Globosae, 10–12.5(–13.5) μm, guttulis 2–4 instructae laeves	Filiformes apice curvatae	Ad terram nudam in olla	[105]
*P. laeterubra*	Laete rubro,	Cylindracei, apice rotundati, 180–200 × 8 μm, 8 sporae		Globosae, guttam 1 magnam oleosam includentes, 10 μm	Filiformes, septatae,ad apicem		[106]
*P. miltina*	Crimson	linear		Globose, 1/1750 of an inch in diameter		On the bare ground, amongst moss, on haills	[107]
*P. multiguttula*	Orange Chrome and cadmium orange	Cylindracei, octospordei, 210–240(–290) × 14–19 μm		Periecte globosae,5–9-guttulatae, 12–13(–14)μm	Non ramosae, septatae, ad apicem crassatae	super terram	[82]
*P. mussooriensis*	Cream-colored to yellow	Cylindric, apex rounded to subtruncate or truncate, 180–213 × 9–13.5 μm		Overlapping in some asci, globose, smooth, 9.7–11.2 μm	Filiform, nonseptate, simple or branched, apex not swollen but strongly curved, 178–218 × 0.7–2 μm	On soil amid mosses	[108]
*P. neotropica*	Pale yellowish-greenish	165–177 × 11–14 μm, 8-spored, not broad at base	Arising from prominent croziers	Globose, with a single large oil globule, smooth-walled, (12–)13–14(–15) μm	Thin, apex swollen, mostly curved in the upper portion but not strongly	On burned wood	[4]
*P. nepalensis*	Yellow	180–218 × 14–18 μm, 8-spored,	Lack regular croziers	12.8–15.5 μm	Expanding, up to 3 μm at their bent to curved apices	On charcoal and burnt soil around a fireplace in a bamboo grove	[86]
*P. orichalcea*	Lutea	Cylindraceis		Globosis, laevibus	Leniter incrassates, flavidia, septatis	On the ground	[83]
*P. pyrophila*	Salmon-pink	Cylindrical, 150–200 × 10–12 μm, 8-spored		Globose, 7–9 μm, 1-seriate, smooth	Filiform, hooked at their apices, often forked	In the spring, on burnt ground	[87]
*P. salmonicolor*	Pale slmon-colored	Cylindric or subcylindric, 275 × 20–24 μm	Not described	Globose, 20 μm	Clavate, reaching a diameter at their apices	On bare ground	[88]
*P. subaurantia*	Orange	Cylindriques		Globuleuses, lisses, uni-ou pluriguttulees, 12–15 × 12–15 μm	Filiformes, largement reeourbees au Sommet	Espece cespiteuse croissant parmi la mousse	[109]
*P. sublaeterubra*	pastel orange	Cylindrical, 254–302 × 15.7–20.9 μm, 8-spored, operculate, becoming narrow towards the base,	distinctly forked base formed by a crozier	spherical, 14.8–17.1 μm, smooth	filiform, slender, septate, branched, curved at apex without enlarged	grows in clusters on sandy grasslands in broad-leaved forests and shrubs	This study
*P. tetraspora*	Possibly orange when fresh	Cylindracei, narrowed below into a short foot, 180 × 17–19 μm, mostly 4-spored when mature, a few with 2 or 5 spores,but forming 8 spores ay first		Globosae to slightly ellipsoid, globule,quite smooth, 15–17 μm,1-guttulatae	Filiformes, numerosae, doubtfully septatae, not enlarged at apex	in terra	[84]

**Table 3 jof-11-00060-t003:** Current list of *Microstoma* species and their diagnostic characteristics. Non-English descriptions from references are not translated to avoid misunderstanding.

**Species**	**Apothecia Color When Fresh**	**Asci**	**Crozier**	**Ascospores**	**Paraphyses**	**Habitat**	**References**
*M. aggregatum*	Roseus	Cylindracei rotundati ad apicem crassitunicati operculati cum operculo laterali stipitati octospori, 214.2–272.0 × 12.8–14.4 μm		Subcrassitunicatae cllipsoideae laeve 24.0–32.0 × 9.6–12.8 μm	Filiformes septatae, simplices vel inferne ramosae superne subincrassatae usque 3 μm	In ligno putrescentis	[110]
*M. apiculosporum*	Orange red	Clavate, suboperculate, 325–335 × 12–14 μm, 8-spored		Ellipsoid, smooth, 25–30 × 9–10 μm, filled with oil droplets, apiculate at both ends of spores, apiculi hemispherical, surrounded with a gelatinous sheath when freshly released	Branched, connected as a net around the asci, filled with many orange granules	On dead sticks of broadleaf tree	[89]
*M. camerunense*	Pinkish	Cylindrical, suboperculate, 245–335 × (6–)7–12 μm, tapering towards the base to a flexuous stalk with thickened walls in the subhymenium		Narrow ellipsoid to fusiform, (13–)14–16(–17) × 3–4(–4.5) μm, smooth, apiculate, at ends, apicual conical	Branched, connected as a net, with subclavate apices	On dead wood	[90]
*M. changchunense*	Light pink	cylindrical, 295–345 × 13.5–19.5 μm, suboperculate, with slightly thick walls		long elliptical, 25.0–36.2 × 11.5–14.5 μm, with a or more guttulate when mature, without appurtenance on the ends, smooth	filiform, septate, branched	scattered on rotten wood in broad-leaved forests	This study
*M. floccosum*	Eleganter coccineo	Cylindraceis		Ellipticis, 20 × 11 μm	filiformibus	Ramos dejectos et terram	[93]
*M. jilinense*	Salmon orange	cylindrical, 247–325 × 10.6–14.2 μm, 8-spored, suboperculate, with slightly thick walls		elliptical to cylindrical, 15.9–24.3 × 9.0–14.9 μm, with a large guttulate when mature, without appurtenance on the ends, smooth	filiform, septate, branched	scattered on rotten wood in mixed coniferous and broad-leaved forests	This study
*M. macrosporum*	Aurantio-cinnabarnae	Cylindro-clavati, 500–560 × 23–26 μm, octospori		Ellipsoideae vel fusiformes, 42–60 × 16–21 μm, smooth, crassituicatae, multi-guttulatae	Filiformes, septatae, ramosae et anastomosantes inter se	Dead stem on the ground in early winter tjrought early spring	[92]
*M. protractum*	Externally bright orange-red, interior, vivid rose-red	Cylindrical, 200–275 × 20–23 μm, 8-spored, operculum lateral		Ellipsoid, to fusoid, 24–45 × 10–14 μm, usually, containing conspicuous globules which vary in size and number	Not flexuous, dichotomously branched several time	On buried sticks and root	[91]
*M. longipilum*	Dull pink to pale orange	Clavate, 275–350 × 10–17.5 μm, operculate with eccentric opercula, thick-walled		Ovoid to ellipsoid without apiculi, (20–)21.9–26.1(–27.5) × 11–12.5 μm, smooth,	Filiform, with apices sometimes swelling or branched irregularly	On rotten wood	[56]
*M. ningshanicum*	Red, orange-red, or light orange-red	Subcylindrical, 8-spored, 449–517 × 16–20.5 μm		Long ellipsoidal to cylindrical, 31–45 × 13–17 μm, non-smooth, without appurtenance on both end		On rotten wood hodden in the ground in a broad-leaved forest	[58]
*M. radicatum*	Hymenuium surface rosy red to red, receptacle surface red, orange-red to light orange-red	Subcylindrical, 8-spored, 390–575 × 15–22 μm		Ellipsoidal, to fusoidal-ellipsoidal, (25–)35–50(–60) × (11–)12.58–20(–22.5) μm, smooth, uniguttulate	Filiform, with apex slightly enlarged	on rotten wood under Larix gmelinii (Rupr.) Rupr. forest	[29]

**Table 4 jof-11-00060-t004:** Current list of part *Sarcoscypha* species and their diagnostic characteristics. Non-English descriptions from references are not translated to avoid misunderstanding.

Species	Apothecia Color When Fresh	Asci	Crozier	Ascospores	Paraphyses	Habitat	References
*S. aestiva*	Ohnive cervine (trochu do oranzova)	Verc. Valcor., uftat. 12–15 μm tl. s 8 jednor.		zaoblene, ellipt., hladke, s 2 velikymi tel., 25–30 μm	velmi hojne, na konci kyjovite ztlustele a oranzove, jodem zelene.		[111]
*S. albovillosa*	Coccineo	Cylindrico, apice truncatis, 300 × 15 μm		Ellipsoideae, guttuam 1 magnam centralem incudentes, episporio sexangulariter retuiculato, 18–21 × 10–12 μm	Filiformes, apice vix hamatae, apice sensim usque 5 μm crassae, jodii ope coerulee decolorates repletae	Ad terram	[112]
*S. austriaca*	Laete cinnabarino, serius coccineo	Cylindraceis, versus apicem rotundatum paullo ampliatis, 437–500 × 22 μm		Ellipsoideis vel orculaeformibus, levibus, 34–38 × 11–14 μm	Filiformibus apice incrassates roseis	In ramis humo tectis	[93]
*S. cerebriforimis*	Bright yellow	Subcylindrical, 290–350 × 12–13 μm,		Rectangular ellipse, 23–28 × 9.5–11.5 μm, slightly rough, with two guttulates	Filiform, 2–3 μm wide	On hardwood	[15]
*S. concatenata*	Extus albis, sericeis, venosis, intus avellaneo-roseis	Cylindricis, 10–12 μm, latis		18–30 μm, longis, 10 μm latis, plerumque ellipticis	Filiformes	In ramo pini	[97]
*S. coccinea*	Scarlet	Cylindrical, 15–17 μm wide		23–33 × 10–14, ellipsoid, Both ends are blunt and round, without depression, smooth, two or more guttulates	Filiform, slightly thicker at the top, branched, septate	On rotten wood	[15]
*S. dawsonensis*	Red or orange	Cylindric, 200–280 μm long		Elliptic, 20 μmlong, 10 μm broad	Filiform, slender, slightly thicker at the top	Among mosses	[113]
*S. dudleyi*	Bright yellow inclining to saffron or orange	cylindrical		Oblong, 0.001 to 0.0012 in. long, 0.0005 to 0.0006 broad,	Filiform, slightly thickened at the tips	Ground and decayed wood	[79]
*S. excelsa*	Carneo-coccineo	Cylindracei, ad apicem obtuse, 250–400 μm longi, 16–20 μm crassi		Anguste ellipsoideae vel potius ellipsoideo-fusiformes, leves, 26–33 × 8–9 μm	Ramosae, ad apicem dilatatae	Ad terram arenosam, ad litora marina	[114]
*S. groenlandica*	Warzchen im unreifen zustande gelbgranlich					auf blattern von saxifraga cernua L. f. ramose	[115]
*S. hongshiensis*	Pure orange	cylindrical, 284–323 × 9.0–9.9 μm, 8-spored, J- in Melzer’s reagent, suboperculate, becoming narrow towards the base, slim, curvaceous.		Cylindrical to rod-shaped, partly both ends truncated, 19.2–27.0 × 8.2–9.7 μm, smooth, with a large guttulate	filiform, septate, branched	scattered on rotten wood in mixed coniferous and broad-leaved forests	This study
*S. hosoyae*	“Grenadine Red” (Ridgway) to “Scarlet” or rarely white	(320–)350–430(–460) × 13–18 μm		22–38(–45) × 9–12 μm, some truncate	Filiform, branching most in lower and upper 1/3, and anastomosing throughout length, do not exceed mature asci, slightly enlarged at tip to spearshaped, rounded, or irregularly lobed	Wet areas of deciduous woods on partially buried twigs/branchlets of angiosperms	[95]
*S. humberiana*	“Grenadine Red” (Ridgway) to “Scarlet”	(230–)270–310(–330) × 10–12.5 μm, tapering, then expanding at base		(16.5–)18–23(–27) × 8.5–10(–12) μm, shallow depression at truncate ends	Generally, evenly filiform with apex slightly enlarged clavate, branching	Wet area on partially buried teigs/branchlets	[95]
*S. javensis*	Hellkarminrot, heller rötlich	Zylindrisch, 230 × 10 μm, achtsporig		Zylindrisch-elliptisch, beidendig meist zbgestumpft, ohne öltropfen, 23 × 8 μm	Fadig unten ein bis zweimal verzweigt, Jod gibt keine Blaufarbunfg	An morschem holze	[94]
*S. jurana*	pulchre-coccineo	Longissimae, operculatae, octosporae, 350–450 μm longae, 15 μmcrassae		Oblong, aut oblong-truncatae, leaves, 24–29 μm longae. 13–14 μm latae	Tenues, dichotomo-ramosae, ramis acutis, iodo caerulescentibus aut virentibus	Ad ramos infossos tiliae	[116]
*S. kecskemetiensis*	pallide sulphurea	Cylindraceis, 120–140 × 8–9 μm, octosporis		Ovatis, levibus, biduttulatis, 12 × 6 μm	Filiformibus, non septatis, apice increassatis	ad terram inter folia decidua et lignula putrida	[117]
*S. knixoniana*	“Spectrum Red” to rose red	(250–)270–350(–420) × 9–13 μm		18–25 × 8–12 μm, with shallow or slightly deeper depressions	Generally simple and filiform, with slightly inflated to clavate, some slightly hooked, branching	Wet areas of deciduous wood on partially buried twigs/branchlets of angiosperms	[95]
*S. korfiana*	“Picric Yellow”, “Lemon Chrome”	220–265(–340) × 9–12 μm,		All shallowly truncate with slight depression at the poles, (14–)16–21 × (7–)8–11 μm	Filiform, gradually tapering at the slightly enlarged, clavate-shaped	on wood	[95]
*S. longitudinalis*	Broen	Subcylindrical, terminally operculate, apex obtuse, 8-spored, 197–359 × 12–14 μm		Broadly fusiform, (18.3–)19.3–21.4(–22.4) × (9.5–)10.7–12.1(–12.8) μm, ornamentation with several longitudinal striae	Filiform, branched, septate, with a rounded end	On unidentified dead branch under broadleaved forest	[44]
*S. macaronesica*	lucido-sanguineo, margine ochraceo-albido	cira 350/13.5 μm (turgescente)		Ellipsoidales, 22–29 × 9–12 μm, lisas,	Multis anastomosis instructae, apice leniter incrassatae	in ramulis dejectis crescentia	[32]
*S. mesocyatha*	“Grenadine Red” to “Scarlet”	(260–)280–330–360) × 10–11(–12.5) μm usually eight ascospores but sometimes only six or senve		(16.5–)21–27(–30) × 9–10.5(–12.5)	Filifirm, with apex slightly enlarged, clavate, banching	Wet area on partially buried teigs/branchlets	[95]
*S. minuta*	Orange to Cadmium Yellow	Cylindrical, 8-spored, 180–250 × 7–10 μm	Without	Broadly ellipsoid, (15–)16–20(–22) × (7–)8–10 μm, smooth, containing a large guttule	Filiform, slightly exceeding the asci, sparsely septate, aoex simple	On fruit shell	[79]
*S. occidentalis*	Luteo-coccineo	Cylindraceis		Obtuse ellipticis, 18 × 9 μm	filiformibus	Ad ramos dejectos	[93]
*S. pseudomelastoma*	schwarzlich-olivenfarbig	Cylindrisch, 180–200 μm		ellipsoid, glatter, arbloser membrane, 12–16 × 6–8 μm	Fadenforming, nacho ben kaum verdickt,gelblich	zwishchen Torfmoospolstern wachsende	[118]
*S. racovitzae*	Disco luteo-aurantio, extus albida	Cylindraceis, 270 × 12–15 μm		Ellipsoideis, 18–20 × 8–9 μm	linearibus rarioribus	In ligno putrescente	[115]
*S. rubrans*	Rouge brillant			Ellipsoide lanceolee, bi-triocellee		Sur les branches moortes	[119]
*S. shennongjiana*	Orange red to red	Suboperculate, 8–spored, 240–292 × (9.9–)10–13.5 μm		Ellipsoid with shallow depression at truncate ends, with two large guttulesand many small ones	subcylindrical	On twigs of a broadleaf tree	[34]
*S. sherriffii*	Levis, aurantioflvum	Cylindracei, 350–400 × 13–15 μm		Ellipsoideae vel oblongae, 25–35 × 10–12 μm, leves	filiformae	Ad ramos putridos dejectos in silva densa	[120]
*S. tatakensis*	Scarlet red	Cylindrical 260–325 × 10–13 μm	Without	Ellipsoid, with shallow depressions at both ends, smooth, (16–)21–27(–30) × 9–11(–14) μm, usually with 2–3 large guttules	Filiform, slightly exceeding the asci, sparsely septate	On dead twigs	[79]
*S. vassiljevae*	Dirty white	Subcylindrical, 290–360 × 9–13 μm		Ellipsoid, 17–25 × 9–13 μm, with a large guttules	Filiform, septate, branched, sometimes connected to a net	On dead twigs	[15]

## Figures and Tables

**Figure 1 jof-11-00060-f001:**
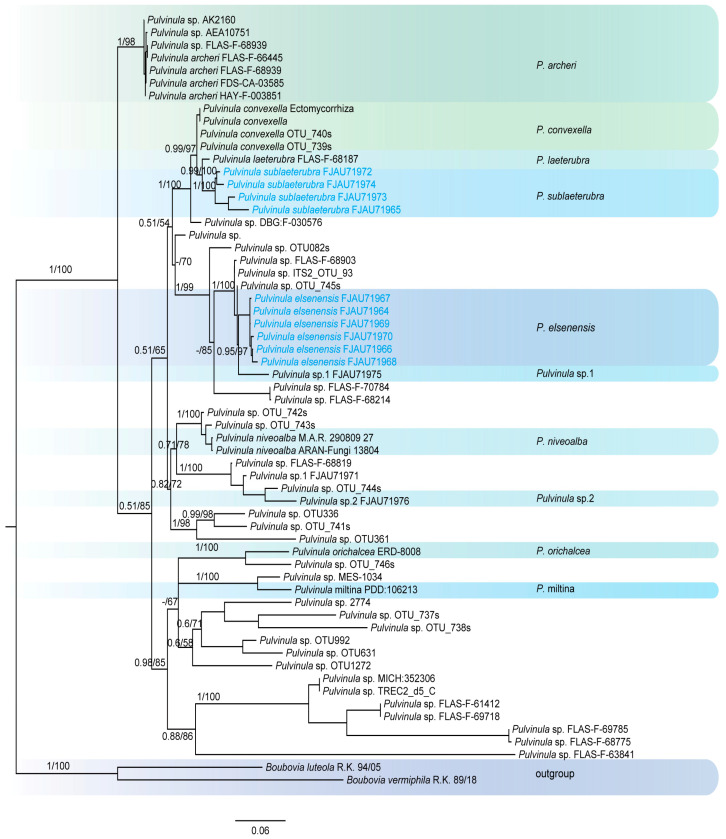
The phylogeny of *Pulvinula* by Bayesian inference based on the ITS and LSU dataset.

**Figure 2 jof-11-00060-f002:**
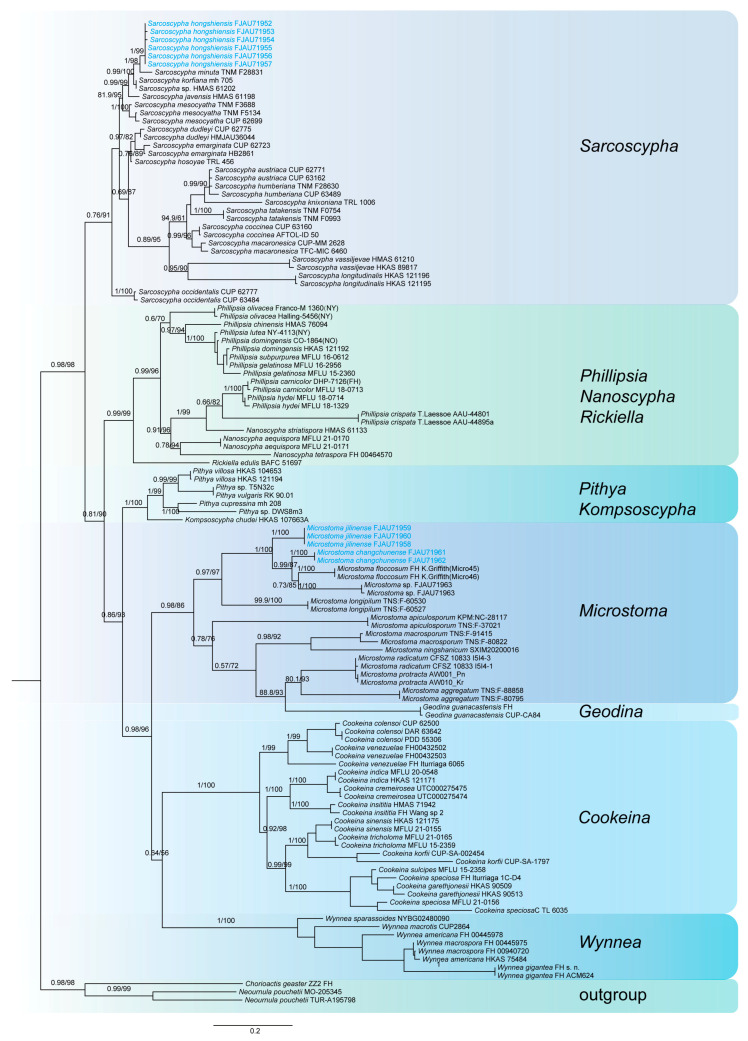
The phylogeny of Sarcoscyphaceae as assessed by Bayesian inference based on the ITS dataset.

**Figure 3 jof-11-00060-f003:**
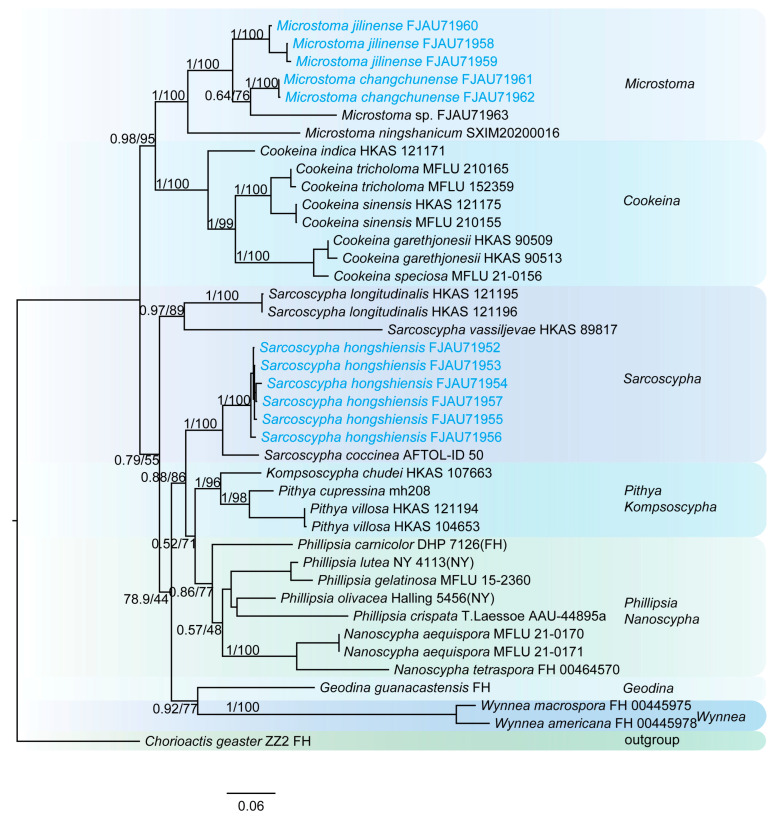
The phylogeny of Sarcoscyphaceae as assessed by Bayesian inference based on the ITS and LSU dataset.

**Figure 4 jof-11-00060-f004:**
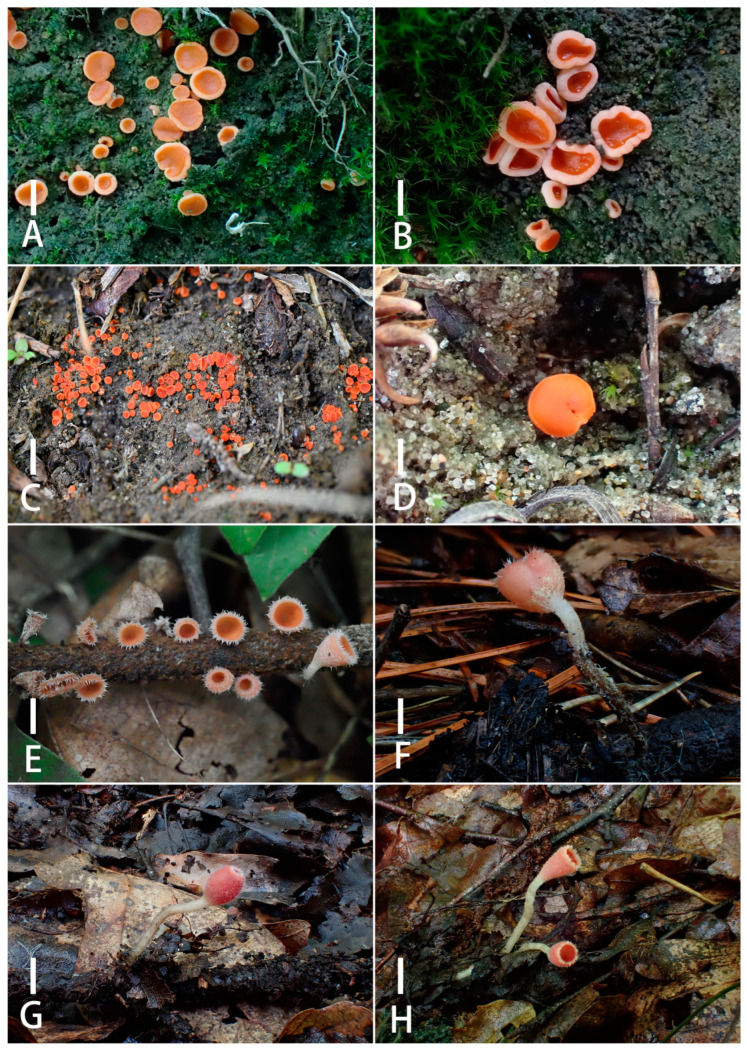
Ascocarps of *Pulvinula elsenensis* (**A**,**B**); *Pulvinula sublaeterubra*; (**C**,**D**) *Microstoma jilinense* (**E**,**F**); *Microstoma changchunense* (**G**,**H**); *Sarcoscypha hongshiensis* (**I**,**J**). Scale bars: (**A**–**D**) = 0.3 cm; (**E**–**J**) = 1 cm. Collection site and collection time: (**A**,**B**): Jilin Province, 2023; (**C**): Inner Mongolia Autonomous Region, 2022, (**D**): Inner Mongolia Autonomous Region, 2023; (**E**): Liaoning Province, 2024, (**F**): Jilin Province, 2024; (**G**, **H**): Jilin Province, 2024; (**I**, **J**): Jilin Province, 2023.

**Figure 5 jof-11-00060-f005:**
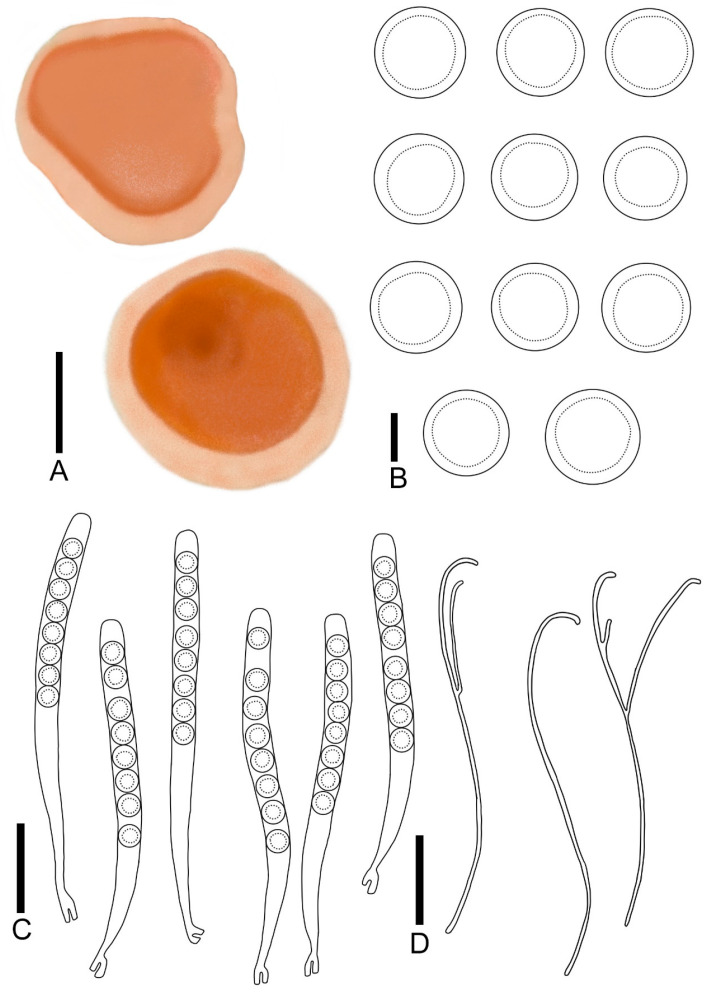
*Pulvinula elsenensis*: (**A**) ascocarps; (**B**) ascospores; (**C**) asci; (**D**) paraphyses. Scale bars: (**A**) = 0.3 cm; (**B**) = 8 μm; (**C**,**D**) = 55 μm.

**Figure 6 jof-11-00060-f006:**
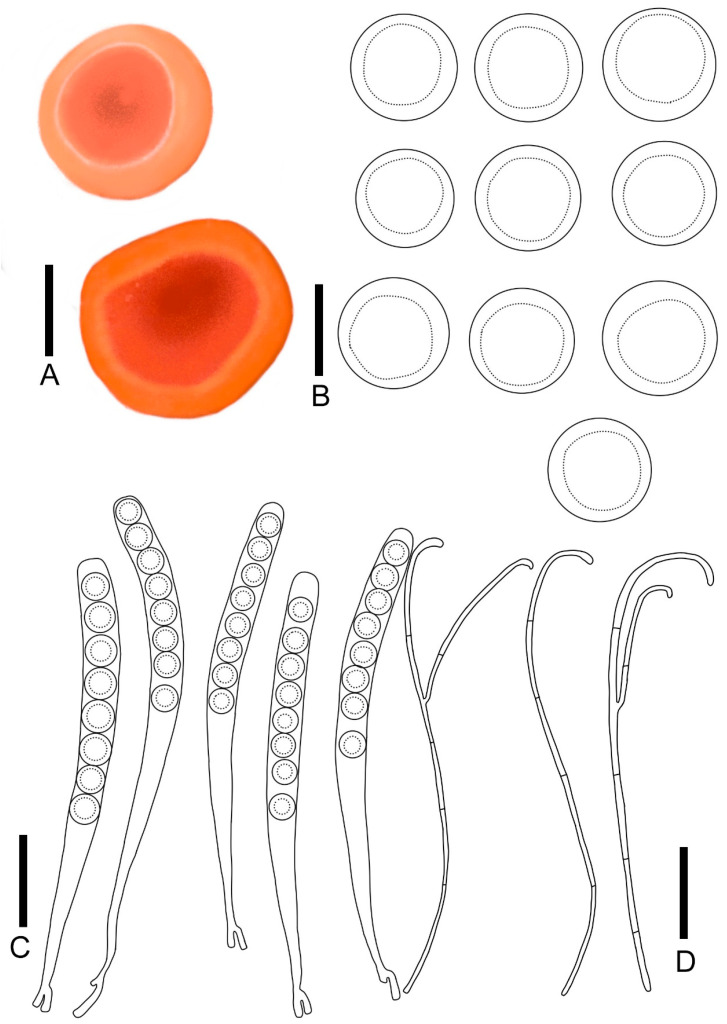
*Pulvinula sublaeterubra*: (**A**) ascocarps; (**B**) ascospores; (**C**) asci; (**D**) paraphyses. Scale bars: (**A**) = 0.3 cm; (**B**) = 15 μm; (**C**,**D**) = 50 μm.

**Figure 7 jof-11-00060-f007:**
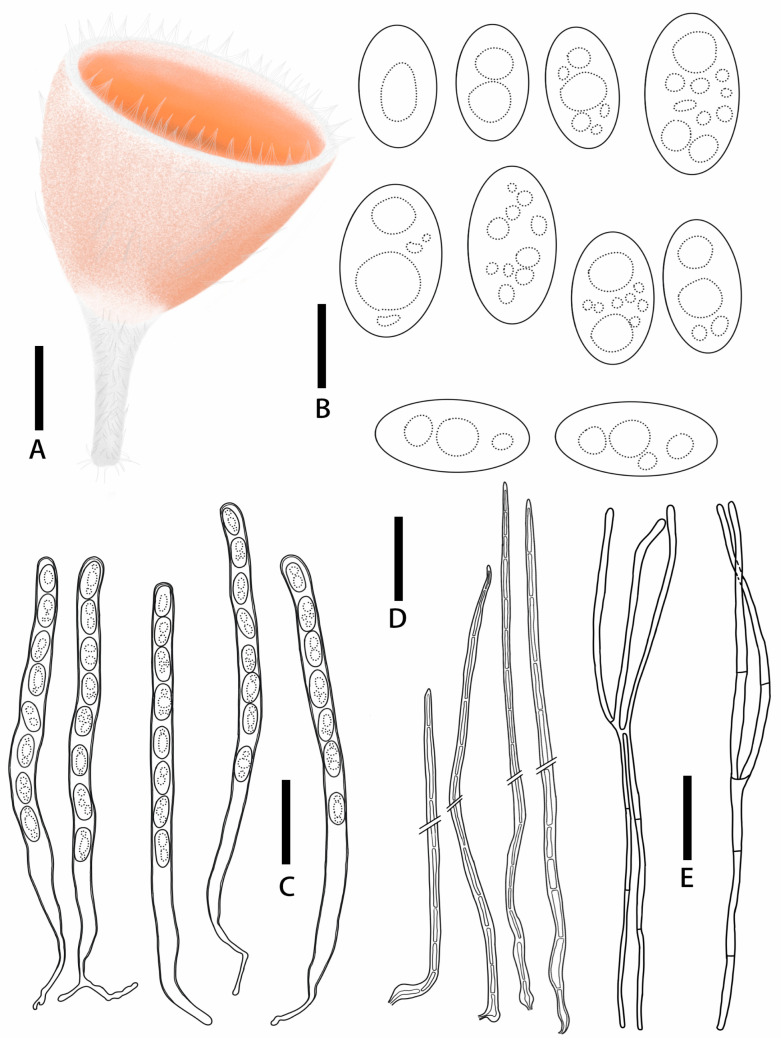
*Microstoma jilinense*: (**A**) ascocarps; (**B**) ascospores; (**C**) asci; (**D**) paraphyses. Scale bars: (**A**) = 0.5 cm; (**B**) = 12 μm; (**C**–**E**) = 55 μm.

**Figure 8 jof-11-00060-f008:**
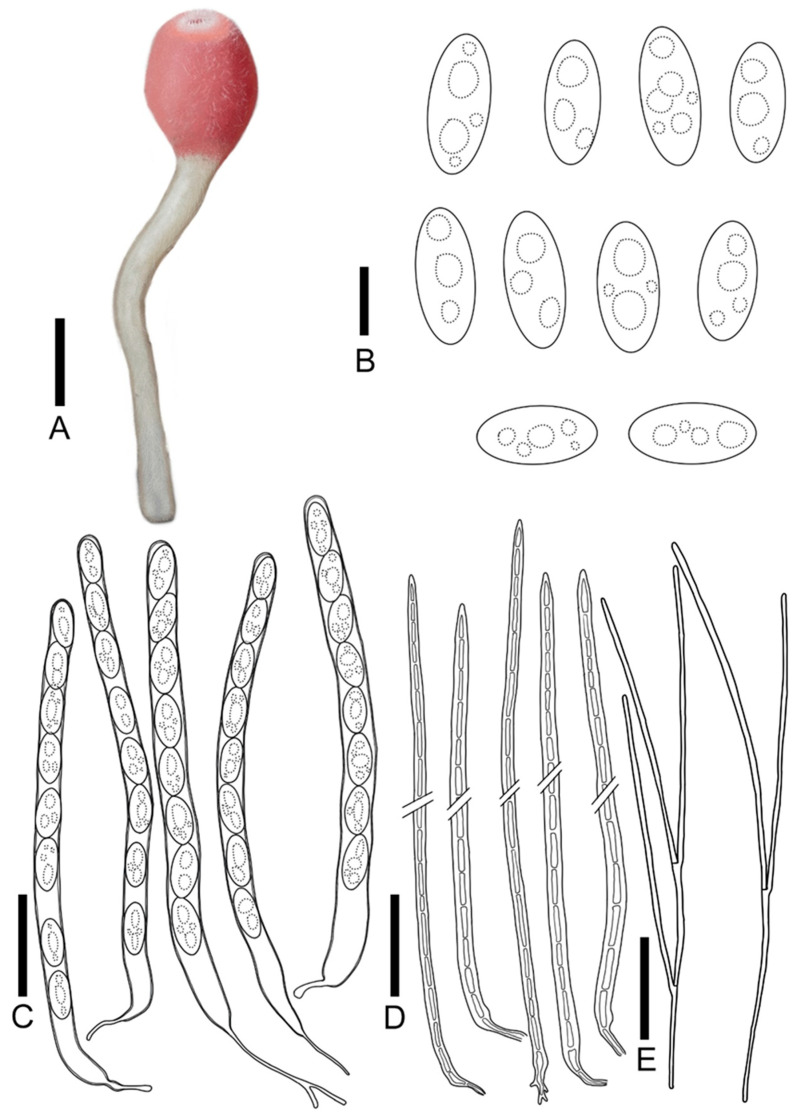
*Microstoma changchunense*: (**A**) ascocarps; (**B**) ascospores; (**C**) asci; (**D**) hairs; (**E**) paraphyses. Scale bars: (**A**) = 0.8 cm; (**B**) = 15 μm; (**C**–**E**) = 55 μm)

**Figure 9 jof-11-00060-f009:**
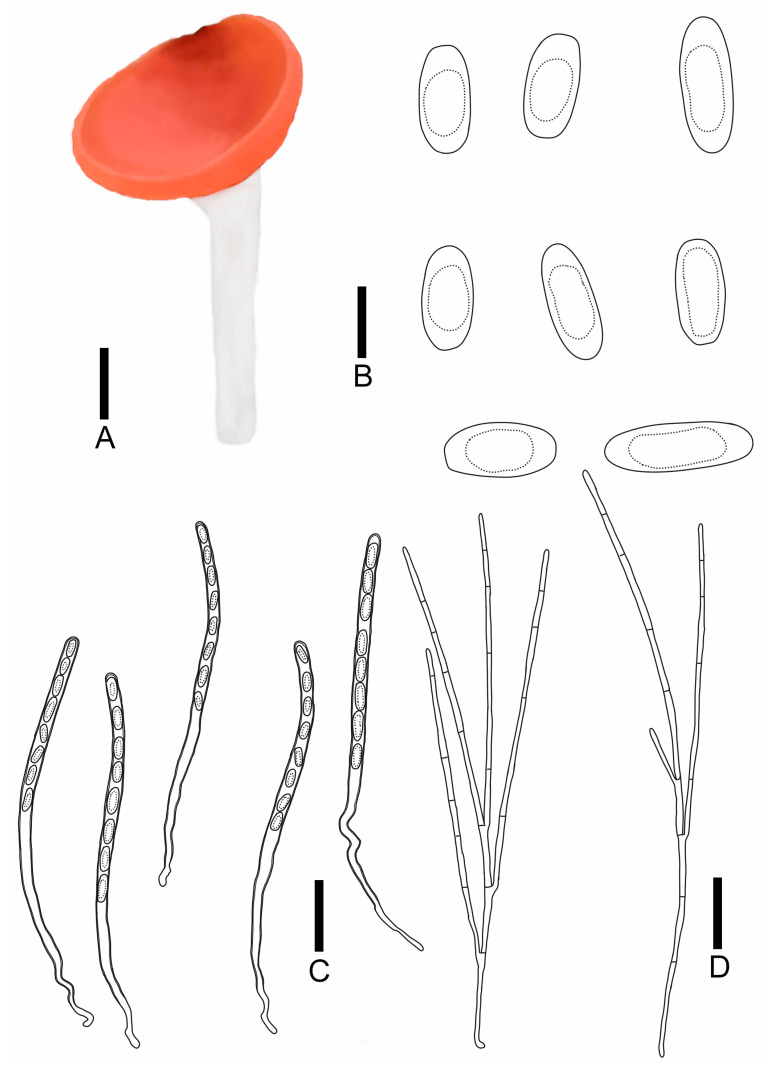
*Sarcoscypha hongshiensis*: (**A**) ascocarps; (**B**) ascospores; (**C**) asci; (**D**) paraphyses. Scale bars: (**A**) = 0.5 cm; (**B**) = 15 μm; (**C**,**D**) = 60 μm.

## Data Availability

All the sequences have been deposited in GenBank (https://www. ncbi.nlm.nih.gov, accessed on 20 November 2024) and MycoBank (https://www.mycobank.org, accessed on 1 December 2024).

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
