# Peer review of "Five New Species of Pezizales from Northeastern China"

_jof, 2025, doi:10.3390/jof11010060_

Round 1
Reviewer 1 Report
The article under review is based on a large amount of factual material obtained by the authors using both traditional mycological and modern molecular genetic methods. The article reports on the discovery by the authors in the North-Eastern part of China (a region poorly studied in mycological terms) of five new species of disc fungi for science, and this makes it interesting and worthy of publication. The manuscript can be accepted for publication in its current form, but taking into account the comments I have made.
The analysis of the manuscript shows that such sections as Abstract and Introduction are well prepared. The introduction discusses in detail the history of the study of taxonomy, phylogenetic relationships of fungi of the order Pezizales. However, I did not find in this part information on whether there were previous studies in this part of China on the study of fungi of this taxon. I believe that this should be noted in the section of the article under consideration. The Materials and Methods section is small in volume, but it gives a complete picture of the technology of morphological and molecular genetic analysis. But, if the number of original sequences obtained by the authors (47 of which 24 ITS, 23 –LSU) is indicated, then there is no data on how many samples of fruiting bodies of fungi of the studied fungi were used in the morphological analysis. There is no description of the study area, of course, this data can be found in the part of the work where the descriptions of species are given, but in the Materials and Methods section it is necessary to provide a short description of the study area. There is also a question regarding Table 1. It is large and takes up several pages, and each page has its own number: Table 2. Cont., Table 3. Cont., Table 4. Cont., Table 5. Cont. Moreover, as one can understand, they are considered as separate tables, this is indicated by the fact that the first table in the Discussion section has the number 6. I find this strange, wouldn’t it be easier to indicate “Table 2, continued” on each page. Results section. The results of a large-scale phylogenetic analysis (a total of 249 sequences!) are presented very briefly. Most of this section of the article is taken up by a detailed description of the newly described species. Their characteristics are detailed and include such points as Diagnosis, Etymology, Type, Habitat, Distribution, Additional specimens examined, Notes. The description of each species is supplemented by high-quality photographs and drawings. However, the location of Figure 4 raises a question: in the manuscript, it precedes the reference to it! Usually it's the other way around - a link and then a drawing. The Discussion section is prepared in a traditional manner; the authors discuss the results of their own morphological and phylogenetic analysis using extensive literature data (the list of references contains 116 sources), and at the end of the discussion, three large tables are presented. Each of them is devoted to fungi of one genus: Pulvinula (Table 6), Microstoma (Table 7), Sarcoscypha (Table 9). They contain and briefly present data on the morphology, anatomy, and ecology of all known species of a given taxon, including newly described ones. However, in the case of these tables, we encounter the same problem that I noted in the example of Table 2: the continuation of one table on each new page has its own number. Let us take, for example, Tables 6 and 7. On pages 19, 20, the continuation of Table 6 is marked as Table 2. Cont., and the continuation of Table 7 on page 22 is marked as Table 8. Cont. The same is true for Table 9 - each part of it on a new page has its own number: from Table 10. Cont. to Table 13. Cont. I think the authors should correct these annoying errors!
The manuscript can be accepted for publication in its current form, but taking into account the comments I have made.
The analysis of the manuscript shows that such sections as Abstract and Introduction are well prepared. The introduction discusses in detail the history of the study of taxonomy, phylogenetic relationships of fungi of the order Pezizales. However, I did not find in this part information on whether there were previous studies in this part of China on the study of fungi of this taxon. I believe that this should be noted in the section of the article under consideration. The Materials and Methods section is small in volume, but it gives a complete picture of the technology of morphological and molecular genetic analysis. But, if the number of original sequences obtained by the authors (47 of which 24 ITS, 23 –LSU) is indicated, then there is no data on how many samples of fruiting bodies of fungi of the studied fungi were used in the morphological analysis. There is no description of the study area, of course, this data can be found in the part of the work where the descriptions of species are given, but in the Materials and Methods section it is necessary to provide a short description of the study area. There is also a question regarding Table 1. It is large and takes up several pages, and each page has its own number: Table 2. Cont., Table 3. Cont., Table 4. Cont., Table 5. Cont. Moreover, as one can understand, they are considered as separate tables, this is indicated by the fact that the first table in the Discussion section has the number 6. I find this strange, wouldn’t it be easier to indicate “Table 2, continued” on each page. Results section. The results of a large-scale phylogenetic analysis (a total of 249 sequences!) are presented very briefly. Most of this section of the article is taken up by a detailed description of the newly described species. Their characteristics are detailed and include such points as Diagnosis, Etymology, Type, Habitat, Distribution, Additional specimens examined, Notes. The description of each species is supplemented by high-quality photographs and drawings. However, the location of Figure 4 raises a question: in the manuscript, it precedes the reference to it! Usually it's the other way around - a link and then a drawing. The Discussion section is prepared in a traditional manner; the authors discuss the results of their own morphological and phylogenetic analysis using extensive literature data (the list of references contains 116 sources), and at the end of the discussion, three large tables are presented. Each of them is devoted to fungi of one genus: Pulvinula (Table 6), Microstoma (Table 7), Sarcoscypha (Table 9). They contain and briefly present data on the morphology, anatomy, and ecology of all known species of a given taxon, including newly described ones. However, in the case of these tables, we encounter the same problem that I noted in the example of Table 2: the continuation of one table on each new page has its own number. Let us take, for example, Tables 6 and 7. On pages 19, 20, the continuation of Table 6 is marked as Table 2. Cont., and the continuation of Table 7 on page 22 is marked as Table 8. Cont. The same is true for Table 9 - each part of it on a new page has its own number: from Table 10. Cont. to Table 13. Cont. I think the authors should correct these annoying errors!
Author Response
Comments 1: The analysis of the manuscript shows that such sections as Abstract and Introduction are well prepared. The introduction discusses in detail the history of the study of taxonomy, phylogenetic relationships of fungi of the order Pezizales. However, I did not find in this part information on whether there were previous studies in this part of China on the study of fungi of this taxon. I believe that this should be noted in the section of the article under consideration
Response 1: Thank you for pointing this out. I gree with this comment. Therefore, in the manuscript I have added previous advances related to the study of Pezizales or Sarcoscyphaceae in northeastern China, but very little research has been done about Pulvinula.
Page 3, paragraph 3, and line 115-134.
Comments 2: The Materials and Methods section is small in volume, but it gives a complete picture of the technology of morphological and molecular genetic analysis. But, if the number of original sequences obtained by the authors (47 of which 24 ITS, 23 –LSU) is indicated, then there is no data on how many samples of fruiting bodies of fungi of the studied fungi were used in the morphological analysis.
Response 2: Thank you very much for raising this question. In the Materials and Methods section, I have now included the number of specimens and ascomata used in the morphological analysis. Additionally, I have used the notation {a/b/c} to indicate that the data on the length, width, and Q value of ascospores are derived from a ascospores of b ascomata from c specimens. Furthermore, I have clearly marked this information in the description sections of the five species.
Page 4, paragraph 1, and line 154-158;
Page 6, paragraph 8, and line 250;
Page 9, paragraph 7, and line 294;
Page 11, paragraph 4, and line 341;
Page 13, paragraph 3, and line 384;
Page 15, paragraph 5, and line 423.
Comments 3: There is no description of the study area, of course, this data can be found in the part of the work where the descriptions of species are given, but in the Materials and Methods section it is necessary to provide a short description of the study area.
Response 3: Thank you for your suggestion regarding the description of China's Northeast region. I believe incorporating this information will enhance the completeness of my manuscript and improve its alignment with the manuscript's title. However, I agree that it would be more appropriate to position this description within the introduction section.
Page 3, paragraph 3, and line 115-130.
Comments 4: There is also a question regarding Table 1. It is large and takes up several pages, and each page has its own number: Table 2. Cont., Table 3. Cont., Table 4. Cont., Table 5. Cont. Moreover, as one can understand, they are considered as separate tables, this is indicated by the fact that the first table in the Discussion section has the number 6. I find this strange, wouldn’t it be easier to indicate “Table 2, continued” on each page.
Response 4: Thank you very much for pointing out the issue with the Tables 1. The discrepancy occurred when I mistakenly labeled the continued tables as "Table 2. Cont.," "Table 3. Cont.," and so on. I have now corrected these errors.
Page 5, paragraph 1, and line 189;
Page 6, paragraph 1, and line 190;
Page 7, paragraph 1, and line 191;
Page 8, paragraph 1, and line 192;
Page 9, paragraph 1, and line 193.
Comments 5: Results section. The results of a large-scale phylogenetic analysis (a total of 249 sequences!) are presented very briefly. Most of this section of the article is taken up by a detailed description of the newly described species. Their characteristics are detailed and include such points as Diagnosis, Etymology, Type, Habitat, Distribution, Additional specimens examined, Notes. The description of each species is supplemented by high-quality photographs and drawings. However, the location of Figure 4 raises a question: in the manuscript, it precedes the reference to it! Usually it's the other way around - a link and then a drawing.
Response 5: Thank you very much for highlighting the issue concerning the photos of the species' habitats in Figure 4 and their locations. When writing the manuscript, I combined them in order to facilitate a direct comparison between the two species within the same genus, making the differences among species more visually apparent.
Comments 6: The Discussion section is prepared in a traditional manner; the authors discuss the results of their own morphological and phylogenetic analysis using extensive literature data (the list of references contains 116 sources), and at the end of the discussion, three large tables are presented. Each of them is devoted to fungi of one genus: Pulvinula (Table 6), Microstoma (Table 7), Sarcoscypha (Table 9). They contain and briefly present data on the morphology, anatomy, and ecology of all known species of a given taxon, including newly described ones. However, in the case of these tables, we encounter the same problem that I noted in the example of Table 2: the continuation of one table on each new page has its own number. Let us take, for example, Tables 6 and 7. On pages 19, 20, the continuation of Table 6 is marked as Table 2. Cont., and the continuation of Table 7 on page 22 is marked as Table 8. Cont. The same is true for Table 9 - each part of it on a new page has its own number: from Table 10. Cont. to Table 13. Cont. I think the authors should correct these annoying errors!
Response 6: Thank you very much for pointing out the issue with the Tables 6, Tables 7, Tables 8 The discrepancy occurred when I mistakenly labeled the continued tables as "Table 2. Cont.," "Table 3. Cont.," and so on. I have now corrected these errors.
Page 19, paragraph 1, and line 508;
Page 20, paragraph 1, and line 509;
Page 22, paragraph 1, and line 512;
Page 23, paragraph 1, and line 515;
Page 24, paragraph 1, and line 516;
Page 25, paragraph 1, and line 517.
Page 26, paragraph 1, and line 518.
Page 24, paragraph 1, and line 516.
Reviewer 2 Report
Chen and Bau described five new species within the order Pezizales in China. The work has merits and it is worth publishing. However, I have some suggestions to improve the Manuscript. So, I recommend major revision.
The first sentence in the Abstract is a bit odd, and not biologically correct. Only species exist in the nature, not taxa. And, there is no need to use “fungal taxon”, we already know that Pezizales is order. Also, I suggest avoiding the term saprophytic and using saprobe instead. So, my suggestion is to rephrase the first sentence in Abstract. (for example. Species belonging to the order Pezizales are primarily saprobes in the nature), and to replace the word saprophytic with the word saprobe in the whole Manuscript.
Introduction
Line 23 I suppose name of the family is Humariaceae not Humaries
Material and Methods.
Table 1. I suggest the row with sequences used in this study to be marked with bold.
Line 153. What was the criteria for selecting members of genera Chorioactis, Neournula and Boubovia as outgroups?
Results.
In these kind of papers usually morphological analyses goes before the phylogenetic ones. I suggest reorganizing result section accordingly.
Figure 4. I would add more information about collection site and sampling period of the ascocarps presented on the Figure.
Author Response
Comments 1: The first sentence in the Abstract is a bit odd, and not biologically correct. Only species exist in the nature, not taxa. And, there is no need to use “fungal taxon”, we already know that Pezizales is order. Also, I suggest avoiding the term saprophytic and using saprobe instead. So, my suggestion is to rephrase the first sentence in Abstract. (for example. Species belonging to the order Pezizales are primarily saprobes in the nature), and to replace the word saprophytic with the word saprobe in the whole Manuscript.
Response 1: Thank you for your valuable feedback. I wholeheartedly agree with your comments. I will rephrase the first sentence in the abstract as you suggested: "Species belonging to the Pezizales are primarily saprobes in nature." In Page 2, paragraph 2, line 61, I believe the term "saprophytic fungi" should be used. "Saprophytic" describes the nutritional mode or lifestyle of an organism, specifically those that obtain nutrients by decomposing dead organic matter. "Saprophytic fungi" refers to fungi that live in a saprophytic manner, acting as decomposers in the ecosystem and absorbing nutrients from organic materials such as animal and plant residues. Furthermore, the majority of species in the Sarcoscyphaceae family are saprophytic fungi found in moist woodlands. These fungi play a crucial role in decomposing plant residues within the natural material cycle. They grow on various plant tissues, including fallen logs at different stages of decay, branches of varying sizes, coniferous tree leaves, and the basal parts of monocotyledonous plants.
Thank you again for your insightful suggestions. They have been extremely helpful in improving the quality of the manuscript.
page 1, paragraph 6, and line 8.
Page 2, paragraph 2, line 61
Comments 2: Line 23 I suppose name of the family is Humariaceae not Humaries
Response 2: Thank you very much for raising this question. I would like to clarify the matter as follows: In 1885, when Boundier published the article Nouvelle classification naturelle des Discomycètes charnus connus généralement sous le nom de Pezizes, the classification system and hierarchical structure of fungi were different from those in use today. Additionally, the suffix for family names was not yet standardized to the current "-aceae." Therefore, I included the word "family" before "Humaries" in the manuscript to clarify that "Humaries" referred to a taxonomic unit at the family level during that time.
page 1, paragraph 9, and line 22-23.
Comments 3: Material and Methods.Table 1. I suggest the row with sequences used in this study to be marked with bold.
Response 3: Thank you for your suggestion regarding Table 1 in the Material and methods section. I fully agree with your recommendation to mark the row with sequences used in this study in bold. I've made the necessary changes to the table to ensure that the row is clearly highlighted.
page 5, page 6, page 7, page 8, and page 9.
Comments 4: Line 153. What was the criteria for selecting members of genera Chorioactis, Neournula and Boubovia as outgroups?
Response 4: Thank you very much for raising this question. The choice of Chorioactis and Neournula as outgroups is based on the relatively close phylogenetic relationship between the family Chorioactidaceae, to which these genera belong, and the Sarcoscyphaceae.
Similarly, Boubovia was selected as an outgroup because, in their 2007 study of the Pyronemataceae, Perry, Hansen, and Pfister found that Boubovia and Pulvinula are sister taxa. Furthermore, the families Ascodesmidaceae and Pulvinulaceae, to which these genera belong, also exhibit a close phylogenetic relationship.
Perry, B. A., Hansen, K., & Pfister, D. H. (2007). A phylogenetic overview of the family Pyronemataceae (Ascomycota, Pezizales). Mycological Research, 111(5), 549-571.
Ekanayaka, A. H., Hyde, K. D., Jones, E. G., & Zhao, Q. I. (2018). Taxonomy and phylogeny of operculate discomycetes: Pezizomycetes. Fungal Diversity, 90, 161-243.
Comments 5: Results. In these kind of papers usually morphological analyses goes before the phylogenetic ones. I suggest reorganizing result section accordingly.
Response 5: Thank you very much for raising this question. I'd like to provide the following explanation for this issue: In this type of article, the conclusion section usually comes before the morphological study. For example, in "Phylogeny and morphology of novel species and new collections related to Sarcoscyphaceae (Pezizales, Ascomycota) from Southwestern China and Thailand", "Four new species of Cystolepiota (Agaricaceae, Agaricales) from northeastern China", "Morphological and phylogenetic analyses reveal five new species of Porotheleaceae (Agaricales, Basidiomycota) from China", and "Morphological reassessment and molecular phylogenetic analyses of Amauroderma s. lat. raised new perspectives in the generic classification of the Ganodermataceae family".
Zeng, M., Gentekaki, E., Hyde, K. D., Zhao, Q., Matočec, N., & Kušan, I. (2023). Phylogeny and morphology of novel species and new collections related to Sarcoscyphaceae (Pezizales, Ascomycota) from Southwestern China and Thailand. Biology, 12(1), 130.
Zhou, X. Y., & Bau, T. (2024). Four new species of Cystolepiota (Agaricaceae, Agaricales) from northeastern China. Frontiers in Microbiology, 15, 1358612.
Na, Q., Zeng, H., Hu, Y., Ding, H., Ke, B., Zeng, Z., . & Ge, Y. (2024). Morphological and phylogenetic analyses reveal five new species of Porotheleaceae (Agaricales, Basidiomycota) from China. MycoKeys, 105, 49.
Costa-Rezende, D. H., Robledo, G. L., Góes-Neto, A., Reck, M. A., Crespo, E., & Drechsler-Santos, E. R. (2017). Morphological reassessment and molecular phylogenetic analyses of Amauroderma s. lat. raised new perspectives in the generic classification of the Ganodermataceae family. Persoonia-Molecular Phylogeny and Evolution of Fungi, 39(1), 254-269.
Comments 6: Figure 4. I would add more information about collection site and sampling period of the ascocarps presented on the Figure.
Response 6: Thank you for your suggestion regarding Figure 4. I agree wholeheartedly with your recommendation to add more information about the collection site and sampling period of the ascocarps presented in the figure. I will promptly add this information to the figure to ensure that it is clear and complete. This will provide valuable context for the data presented and enhance the overall understanding of the research. Thank you again for your input. Your insights are highly appreciated and help to improve the quality of the manuscript.
page 6, paragraph 1, and line 227-229.
Round 2
Reviewer 2 Report
I agree with changes authors in revised version of the Manuscript. Also, I am satisfied how they responded to the reviewers' comments. Hence, I agree with the publication.
Accept